# Exploring Diffusion Time-steps for Unsupervised Representation Learning

**Zhongqi Yue[1], Jiankun Wang[1], Qianru Sun[2], Lei Ji[3], Eric I-Chao Chang[3], Hanwang Zhang[1,4]**

[1]Nanyang Technological University, [2]Singapore Management University,[3]Microsoft Research Asia,[4]Skywork AI

zhongqi.yue@ntu.edu.sg, jiankun001@e.ntu.edu.sg, qianrusun@smu.edu.sg,
leiji@microsoft.com, eric.i.chang@outlook.com, hanwangzhang@ntu.edu.sg

## Abstract

Representation learning is all about discovering the hidden modular attributes that generate the data faithfully. We explore the potential of Denoising Diffusion Probabilistic Model (DM) in unsupervised learning of the modular attributes. We build a theoretical framework that connects the diffusion time-steps and the hidden attributes, which serves as an effective inductive bias for unsupervised learning. Specifically, the forward diffusion process incrementally adds Gaussian noise to samples at each time-step, which essentially collapses different samples into similar ones by losing attributes, *e.g.*, fine-grained attributes such as texture are lost with less noise added (*i.e.*, early time-steps), while coarse-grained ones such as shape are lost by adding more noise (*i.e.*, late time-steps). To disentangle the modular attributes, at each time-step $t$, we learn a $t$-specific feature to compensate for the newly lost attribute, and the set of all $\{1, \ldots, t\}$-specific features, corresponding to the cumulative set of lost attributes, are trained to make up for the reconstruction error of a pre-trained DM at time-step $t$. On CelebA, FFHQ, and Bedroom datasets, the learned feature significantly improves attribute classification and enables faithful counterfactual generation, *e.g.*, interpolating only one specified attribute between two images, validating the disentanglement quality. Codes are in `https://github.com/yue-zhongqi/diti`.

## 1 Introduction

A good feature representation should faithfully capture the underlying generative attributes in a compact and modular vector space Bengio et al. (2013), enabling not only sample inference (*e.g.*, image classification) but also counterfactual generation Besserve et al. (2020) (*e.g.*, image synthesis of unseen attribute combinations). Over the past decade, discriminative training has been the feature learning mainstream with exceptional performance in inference tasks He et al. (2016; 2020). However, it hardly achieves faithful generation due to the deliberate discard of certain attributes, *e.g.*, class-irrelevant ones in supervised learning or augmentation-related ones in contrastive learning.

On the other hand, generative Denoising Diffusion Probabilistic Model (DM) Sohl-Dickstein et al. (2015); Song et al. (2021) can retain all the underlying attributes for faithful generation Dhariwal & Nichol (2021), or even extrapolate novel attribute combinations by textual control Rombach et al. (2022) (*e.g.*, "Teddy bear skating in Time Square"), outperforming other generative models like VAE Kingma & Welling (2014) and GAN Creswell et al. (2018). This implies that DM effectively captures the modularity of hidden attributes Yue et al. (2021). However, as DM's formulation has no explicit encoders that transform samples into feature vectors, the conventional encoder-decoder feature learning paradigm via reconstruction Higgins et al. (2017) is no longer applicable.

Can we extract DM's knowledge about the modular attributes as a compact feature vector? To answer the question, we start by revealing a natural connection between the diffusion time-steps and the hidden attributes. In the forward diffusion process, given an initial sample $\mathbf{x}_0$, a small amount of Gaussian noise $\mathcal{N}(\mathbf{0}, \mathbf{I})$ is incrementally added to the sample across $T$ time-steps, resulting in a sequence of noisy samples $\mathbf{x}_1, \ldots, \mathbf{x}_T$. In particular, each $\mathbf{x}_t$ adheres to its noisy sample distribution $q(\mathbf{x}_t|\mathbf{x}_0)$, whose mean and covariance matrix $\sigma^2\mathbf{I}$ is denoted as a dot and a circle with radius $\sigma$ in Figure 1a, respectively. As $t$ increases, $q(\mathbf{x}_t|\mathbf{x}_0)$ progressively collapses towards a pure noise by

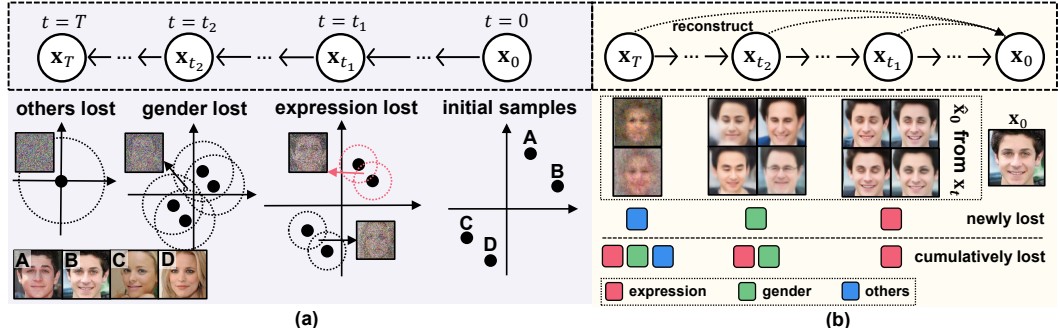

Figure 1: (a) Illustration of attribute loss as time-step $t$ increases in the forward diffusion process. The two axes depict a two-dimensional sample space. (b) DM reconstructed $\mathbf{x}_0$, denoted as $\hat{\mathbf{x}}_0$, from randomly sampled $\mathbf{x}_t$ at various $t$. DM is pre-trained on CelebA, from where $\mathbf{x}_0$ is drawn.

reducing mean (*i.e.*, dots moving closer to the origin), and increasing $\sigma$ (*i.e.*, larger radii), which enlarges the overlapping area (OVL) of the probability density function of noisy sample distributions. In particular, we theoretically show in Section 4.1 that OVL is correlated with DM's ambiguous reconstruction of different samples, *e.g.*, due to the large overlap between the red circles at $t_1$, noisy samples drawn from the two distributions are reconstructed ambiguously as either A or B, losing the attribute "expression"[1] that distinguishes them. As $t$ increases, more noisy sample distributions exhibit significant overlaps, inducing additional attribute loss, *e.g.*, further losing "gender" at $t_2$ that confuses the identities of A/B and C/D. Eventually, with a large enough $T$, $\mathbf{x}_T$ becomes pure noise and all attributes are lost—the reconstruction can be any sample.

This intrinsic connection between diffusion time-steps and modular attributes can act as an effective inductive bias for unsupervised learning. Specifically, DM training can be viewed as learning to reconstruct $\mathbf{x}_0$ given $\mathbf{x}_t$ at each $t$ (Section 3.2). However, due to the aforementioned attribute loss, perfect DM reconstruction is impossible, *e.g.*, in Figure 1b, reconstructions exhibit variations in the cumulatively lost attributes. Hence, by contraposition, if a feature enables perfect reconstruction, it must capture the cumulatively lost attributes. This motivates us to learn an *expanding* set of features to supplement the *expanding* lost attributes as $t$ increases. Specifically, we first map $\mathbf{x}_0$ to a feature $\mathbf{z} = f(\mathbf{x}_0)$ using an encoder $f$. Then we partition $\mathbf{z}$ into $T$ disjoint subsets $\{\mathbf{z}_i\}_{i=1}^{T}$. At each $t$, we optimize $f$ so that $\{\mathbf{z}_i\}_{i=1}^{t}$ compensates for a pre-trained DM's reconstruction error. Intuitively by induction, starting from $t = 0$ with no lost attribute; if $\{\mathbf{z}_i\}_{i=1}^{t}$ captures the cumulatively lost attributes till $t$, $\mathbf{z}_{t+1}$ must capture the newly lost attribute to enable perfect reconstruction at $t + 1$; until $t = T - 1$, $\{\mathbf{z}_i\}_{i=1}^{T-1} \cup \mathbf{z}_T$ learns all hidden attributes in a compact and modular vector space.

We term our approach as **DiTi** to highlight our leverage of Diffusion Time-step as an inductive bias for feature learning. We summarize the paper structure and our contributions below:

- In Section 3.1, we formalize the notion of good feature with a definition of disentangled representation Higgins et al. (2018) and provide a preliminary introduction to DM in Section 3.2.

- In Section 4.1, we build a theoretical framework that connects diffusion time-steps and the hidden modular attributes, which motivates a simple and practical approach for unsupervised disentangled representation learning, discussed in Section 4.2.

- In Section 5, by extensive experiments on CelebA Liu et al. (2015), FFHQ Karras et al. (2019) and Bedroom Yu et al. (2015), our DiTi feature brings significant improvements in attribute inference accuracy and enables counterfactual generation, verifying its disentanglement quality.

## 2 RELATED WORKS

**DM for Representation Learning**. There are three main approaches: 1) DDIM Song et al. (2020) inversion aims to find a latent $\mathbf{x}_T$ as feature, by DDIM sampling with the learned DM, can reconstruct the image $\mathbf{x}_0$. However, this process is time-consuming, and the resulting latent is difficult to interpret Hertz et al. (2022). 2) Kwon et al. (2022) uses the bottleneck feature of the U-Net at all time-steps. However, the feature is not compact and difficult for downstream leverage. 3)

---

[1]Attribute name is illustrative. Our method uses no attribute supervision, nor explicitly names them.

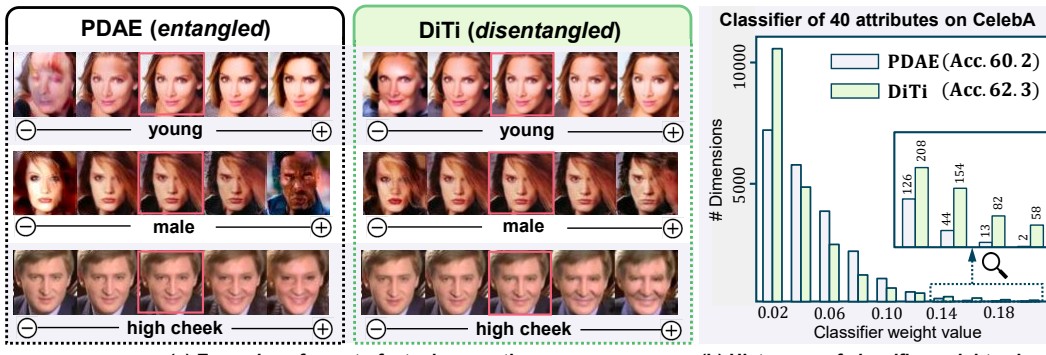

Figure 2: (a) Counterfactual generations on CelebA by manipulating 16 out of 512 feature dimensions (*i.e.*, simulating the edit of a single $\mathbf{z}_i$). A disentangled representation enables editing a single attribute (*e.g.*, gender) without affecting others (*e.g.*, lighting) and promotes faithful extrapolation (*e.g.*, no artifacts). (b) Histogram of the classifier weight value. More dimensions of DiTi weights are closed to 1 and 0 (explanations in the text).

The closest to our works are Preechakul et al. (2022); Zhang et al. (2022). However, they learn a *time-step-independent* $\mathbf{z}$ to supplement DM reconstruction error. Without explicit design to enforce modularity, the learned feature entangles all the attributes. At time-step $t$, it leaks information about attributes lost at a later time-step (*i.e.*, $t + 1, \ldots, T$). Hence, instead of learning the lost attribute $\mathbf{z}_t$, the reconstruction may digress to use the spurious correlation between leaked information and lost attribute. In Section 5, we validate this deficiency by observing failed counterfactual generation and lower attribute prediction accuracy.

**Disentangled Representation** separates the modular hidden attributes that generate data, which can be formally defined from a group-theoretical view Higgins et al. (2018). Our work focuses on the unsupervised learning setting. Existing methods typically learn a VAE Burgess et al. (2017) or GAN Jeon et al. (2021) with information bottleneck. Yet they lack explicit inductive bias that is necessary for unsupervised disentanglement Locatello et al. (2019). Our work reveals that the connection between hidden attributes and diffusion time-steps is an effective inductive bias towards disentanglement (Section 4.1).

## 3 PRELIMINARIES

### 3.1 DISENTANGLED REPRESENTATION

A good feature can be formally defined by the classic notion of disentangled representation. We aim to provide a preliminary introduction below, and refer interesting readers to Higgins et al. (2018); Wang et al. (2021) for more formal details. Each sample $\mathbf{x} \in \mathcal{X}$ in the image space $\mathcal{X}$ is generated from a hidden attribute vector $\mathbf{z} \in \mathcal{Z}$ in the vector space $\mathcal{Z}$ through an injective mapping $\Phi : \mathcal{Z} \to \mathcal{X}$ (*i.e.*, different samples are generated by different attributes). In particular, $\mathcal{Z}$ can be decomposed into the Cartesian product of $N$ subspaces $\mathcal{Z} = \mathcal{Z}_1 \times \ldots \times \mathcal{Z}_N$, where each $\mathcal{Z}_i$ is called a *modular* attribute, as its value can be locally intervened without affecting other modular attributes (*e.g.*, changing "expression" without affecting "gender" or "age"). A disentangled representation is a mapping $f : \mathcal{X} \to \mathcal{Z}$ that inverses $\Phi$, *i.e.*, given each image $\mathbf{x}$, $f(\mathbf{x})$ recovers the modular attributes as a feature vector $\mathbf{z} = [\mathbf{z}_1, \ldots, \mathbf{z}_N]$, which possesses desired properties of a good feature in the following common views:

**Counterfactual generation**. A counterfactual image of $\mathbf{x}$ by intervening the $i$-th modular attribute with $g_i$ (*e.g.*, *aging*) is denoted as $g_i \cdot \mathbf{x} \in \mathcal{X}$ (*e.g.*, an *older* $\mathbf{x}$). Specifically, $g_i \cdot \mathbf{x}$ is generated by first mapping $\mathbf{x}$ to $\mathbf{z}$ by $f$, editing $\mathbf{z}_i$ according to $g_i$ without changing other attributes, and finally feeding the modified $\mathbf{z}$ to $\Phi$. In fact, $g_i \cdot \mathbf{x}$ is formally known as group action Judson (1994). Figure 2 shows counterfactual images generated by editing three attributes.

**Sample inference**. A modular feature facilitates robust inference in downstream tasks about *any* subset of $\{\mathbf{z}_i\}_{i=1}^{N}$, *e.g.*, a linear classifier can eliminate task-irrelevant environmental attributes by assigning larger values to the dimensions corresponding to $\mathbf{z}_i$ and 0 to other dimensions. In Figure 2, by explicitly disentangling the modular attributes (Section 4), our DiTi achieves higher accuracy in

attribute prediction than PDAE Zhang et al. (2022) (without disentangling), while more dimensions of our classifier weight are closed to $1$ (corresponding to $\mathbf{z}_i$) and $0$ (environmental attributes).

## 3.2 Denoising Diffusion Probabilistic Model

Denoising Diffusion Probabilistic Model Ho et al. (2020) (DM) is a latent variable model with a fixed *forward process*, and a learnable *reverse process*. Details are given below.

**Forward Process**. It is fixed to a Markov chain that incrementally adds Gaussian noise to each sample $\mathbf{x}_0$ in $T$ time-steps, producing a sequence of noisy samples $\mathbf{x}_1, \ldots, \mathbf{x}_T$ as latents of the same dimensionality as the original sample $\mathbf{x}_0$. Given $\mathbf{x}_0$ and a variance schedule $\beta_1, \ldots, \beta_T$ (*i.e.*, how much noise is added at each time-step), each $\mathbf{x}_t$ adheres to the following noisy sample distribution:

$$q(\mathbf{x}_t|\mathbf{x}_0) = \mathcal{N}(\mathbf{x}_t; \sqrt{\bar{\alpha}_t}\mathbf{x}_0, (1 - \bar{\alpha}_t)\mathbf{I}), \quad \text{where} \ \ \alpha_t := 1 - \beta_t, \ \bar{\alpha}_t := \prod_{s=1}^{t} \alpha_s. \tag{1}$$

**Reverse Process**. It corresponds to a learned Gaussian transition $p_\theta(\mathbf{x}_{t-1}|\mathbf{x}_t)$ parameterized by $\theta$, starting at $p(\mathbf{x}_T) := \mathcal{N}(\mathbf{x}_T; \mathbf{0}, \mathbf{I})$. Each $p_\theta(\mathbf{x}_{t-1}|\mathbf{x}_t)$ is computed in two steps: 1) Reconstruct $\mathbf{x}_0$ from $\mathbf{x}_t$ with $u_\theta(\mathbf{x}_t, t)$, where $u_\theta$ is a learnable U-Net Ronneberger et al. (2015). 2) Compute $q(\mathbf{x}_{t-1}|\mathbf{x}_t, u_\theta(\mathbf{x}_t, t))$, which has a closed-form solution given in Appendix. Training is performed by minimizing the reconstruction error made by $u_\theta(\mathbf{x}_t, t)$ at each time-step $t$:

$$\mathcal{L}_{DM} = \mathop{\mathbb{E}}_{t, \mathbf{x}_0, \epsilon} \left[ \frac{\bar{\alpha}_t}{1 - \bar{\alpha}_t} \|\mathbf{x}_0 - u_\theta(\sqrt{\bar{\alpha}_t}\mathbf{x}_0 + \sqrt{1 - \bar{\alpha}_t}\epsilon, t)\|^2 \right], \tag{2}$$

where $\epsilon \sim \mathcal{N}(\mathbf{0}, \mathbf{I})$ is a random noise. Note that we formulate the U-Net to predict $\mathbf{x}_0$ (equivalent to the $\epsilon$-formulation in Ho et al. (2020)). As DM has no explicit feature encoder, we learn one to capture the modular attributes by leveraging the inductive bias in the diffusion time-steps.

## 4 Method

### 4.1 Theory

We reveal that in the forward diffusion process, fine-grained attributes are lost at an earlier time-step compared to coarse-grained ones, where the granularity is defined by the pixel-level changes when altering an attribute. We first formalize the notion of attribute loss, *i.e.*, when DM fails to distinguish the noisy samples drawn from two overlapping noisy sample distributions $q(\mathbf{x}_t|\mathbf{x}_0)$ and $q(\mathbf{y}_t|\mathbf{y}_0)$.

**Definition**. (Attribute Loss) *Given a $\theta$-parameterized DM optimized by Eq. 2, we say that attribute $\mathcal{Z}_i$ is lost with degree $\tau$ at $t$ when $\mathbb{E}_{\mathbf{x}_0 \in \mathcal{X}} [\text{Err}(\mathbf{x}_0, \mathbf{y}_0 = g_i \cdot \mathbf{x}_0, t)] \geq \tau$, where $\text{Err}(\mathbf{x}_0, \mathbf{y}_0, t) :=$*

$$\frac{1}{2} \left[ \mathop{\mathbb{E}}_{q(\mathbf{x}_t|\mathbf{x}_0)} [\mathbb{1}(\|\hat{\mathbf{x}}_0 - \mathbf{x}_0\| > \|\hat{\mathbf{x}}_0 - \mathbf{y}_0\|)] + \mathop{\mathbb{E}}_{q(\mathbf{y}_t|\mathbf{y}_0)} [\mathbb{1}(\|\hat{\mathbf{y}}_0 - \mathbf{x}_0\| < \|\hat{\mathbf{y}}_0 - \mathbf{y}_0\|)] \right], \tag{3}$$

*with $\mathbb{1}(\cdot)$ denoting the indicator function and $\hat{\mathbf{x}}_0 = u_\theta(\mathbf{x}_t), \hat{\mathbf{y}}_0 = u_\theta(\mathbf{y}_t)$ denoting the DM reconstructed samples from $\mathbf{x}_t, \mathbf{y}_t$, respectively.*

Intuitively, $\text{Err}(\mathbf{x}_0, \mathbf{y}_0, t)$ measures attribute loss degree by how likely a DM falsely reconstructs $\mathbf{x}_t$ drawn from $q(\mathbf{x}_t|\mathbf{x}_0)$ closer to $\mathbf{y}_0$ instead of $\mathbf{x}_0$ (vice versa). Hence, when $\mathbf{x}_0$ and $\mathbf{y}_0$ differ in attribute $\mathbf{z}_i$ modified by $g_i$, a larger $\text{Err}(\mathbf{x}_0, \mathbf{y}_0, t)$ means that the attribute is more likely lost.

**Theorem**. (Attribute Loss and Time-step) *1) For each $\mathcal{Z}_i$, there exists a smallest time-step $t(\mathcal{Z}_i)$, such that $\mathcal{Z}_i$ is lost with degree $\tau$ at each $t \in \{t(\mathcal{Z}_i), \ldots, T\}$. 2) $\exists\{\beta_i\}_{i=1}^T$ such that $t(\mathcal{Z}_i) > t(\mathcal{Z}_j)$ whenever $\|\mathbf{x}_0 - g_i \cdot \mathbf{x}_0\|$ is first-order stochastic dominant over $\|\mathbf{x}_0 - g_j \cdot \mathbf{x}_0\|$ with $\mathbf{x}_0 \sim \mathcal{X}$ uniformly.*

Intuitively, the first part of the theorem states that a lost attribute will not be regained as time-step $t$ increases, and there is a time-step $t(\mathcal{Z}_i)$ when $\mathcal{Z}_i$ becomes lost (with degree $tau$) for the first time. The second part states that when $\|\mathbf{x}_0 - g_i \cdot \mathbf{x}_0\|$ is more likely to take on a larger value than $\|\mathbf{x}_0 - g_j \cdot \mathbf{x}_0\|$, the attribute $\mathcal{Z}_i$ (*i.e.*, a coarse-grained attribute) is lost at a larger time-step compared to $\mathcal{Z}_j$ (*i.e.*, a fine-grained attribute). We have the following proof sketch:

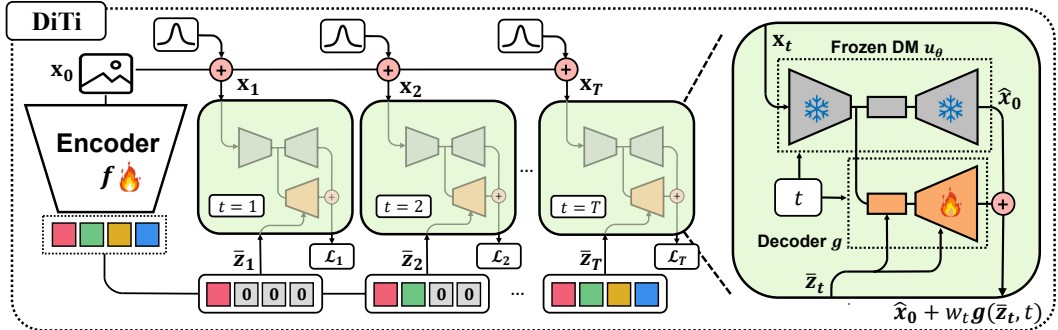

Figure 3: Illustration of our DiTi. We break down Eq. 5 at each time-step. On the right, we show the detailed network design, where $\hat{\mathbf{x}}_0$ denotes the reconstructed $\mathbf{x}_0$ by the pre-trained DM.

*1) Increasing $t$ induces further attribute loss.* We show in Appendix that $\mathrm{Err}(\mathbf{x}_0, \mathbf{y}_0, t) = \frac{1}{2}\mathrm{OVL}\left(q(\mathbf{x}_t|\mathbf{x}_0), q(\mathbf{y}_t|\mathbf{y}_0)\right)$ for an optimal DM, where OVL is the overlapping coefficient Inman & Bradley Jr (1989), *i.e.*, overlapping area of probability density functions (PDFs). In particular,

$$\mathrm{Err}(\mathbf{x}_0, \mathbf{y}_0, t) = \frac{1}{2}\mathrm{OVL}\left(q(\mathbf{x}_t|\mathbf{x}_0), q(\mathbf{y}_t|\mathbf{y}_0)\right) = \frac{1}{2}\left[1 - \mathrm{erf}\left(\frac{\|\sqrt{\bar{\alpha}_t}(\mathbf{y}_0 - \mathbf{x}_0)\|}{2\sqrt{2(1 - \bar{\alpha}_t)}}\right)\right]. \quad (4)$$

As $\bar{\alpha}_t$ decreases with an increasing $t$ from Eq. 1, and the error function $\mathrm{erf}(\cdot)$ is strictly increasing, $\mathrm{Err}(\mathbf{x}_0, \mathbf{y}_0, t)$ is strictly increasing in $t$ given any $\mathbf{x}_0, \mathbf{y}_0$. Hence $\mathbb{E}_{\mathbf{x}_0 \in \mathcal{X}}\left[\mathrm{Err}(\mathbf{x}_0, \mathbf{y}_0 = g_i \cdot \mathbf{x}_0, t)\right] \geq \mathbb{E}_{\mathbf{x}_0 \in \mathcal{X}}\left[\mathrm{Err}(\mathbf{x}_0, \mathbf{y}_0 = g_i \cdot \mathbf{x}_0, t(\mathcal{Z}_i))\right]$ for every $t \geq t(\mathcal{Z}_i)$, which completes the proof.

*2) Coarse-grained attributes are lost at a larger $t$.* Given that $\mathrm{erf}(\cdot)$ is strictly increasing and $\|\mathbf{x}_0 - g_i \cdot \mathbf{x}_0\|$ is first-order stochastic dominant over $\|\mathbf{x}_0 - g_j \cdot \mathbf{x}_0\|$, we have $\mathbb{E}_{\mathbf{x}_0 \in \mathcal{X}}\left[\mathrm{Err}(\mathbf{x}_0, g_i \cdot \mathbf{x}_0, t)\right] > \mathbb{E}_{\mathbf{x}_0 \in \mathcal{X}}\left[\mathrm{Err}(\mathbf{x}_0, g_j \cdot \mathbf{x}_0, t)\right]$ at every time-step $t$ using Eq. 4. Hence $t(\mathcal{Z}_i) > t(\mathcal{Z}_j)$ under any variance schedule $\{\beta_i\}_{i=1}^T$ such that $\mathcal{Z}_i$ is *not* lost at $t(\mathcal{Z}_j)$, completing the proof. Note that in practice, DM leverages a large $T$ with small $\{\beta_i\}_{i=1}^T$. This ensures that $\mathrm{Err}(\mathbf{x}_0, \mathbf{y}_0, t)$ slows increases with $t$ according to Eq. 4. Hence empirically, this theorem tends to hold.

## 4.2 PROPOSED APPROACH

The previous theoretical analysis leads to two interesting groundings: 1) Attribute loss is linked with the error of reconstructing $\mathbf{x}_0$ from the noisy $\mathbf{x}_t$. 2) Increasing time-step $t$ in forward diffusion process causes an expanding cumulative loss of attributes, with fine-grained attributes lost at a smaller $t$ and coarse-grained ones lost at a larger $t$. They serve as an effective inductive bias for unsupervised learning: 1) making up for the reconstruction error can retrieve lost attributes, 2) an expanding set of features captures the expanding cumulatively lost attributes. This motivates our approach below.

As illustrated in Figure 3, our model includes a frozen DM U-Net $u_\theta$ pre-trained on $\mathcal{D}$ using Eq. 2 until convergence, a trainable encoder $f$ (trained to become a disentangled representation) that maps each image (without noise) to a $d$-dimensional feature $\mathbf{z} \in \mathbb{R}^d$, and a trainable decoder $g$ that maps $\mathbf{z}$ back to image space given $t$. In training, given $\mathbf{z} = f(\mathbf{x}_0)$, we partition it into $T$ disjoint subsets $\{\mathbf{z}_i\}_{i=1}^T$. At each time-step $t$, we construct a new feature $\bar{\mathbf{z}}_t = [\mathbf{z}_1, \ldots, \mathbf{z}_t, \mathbf{0}, \ldots, \mathbf{0}]$, which is only about $\{\mathbf{z}_i\}_{i=1}^t$ by masking $\mathbf{z}_{t+1}, \ldots, \mathbf{z}_T$ as $\mathbf{0}$. Our training loss requires $\bar{\mathbf{z}}_t$ to compensate for the reconstruction error made by $u_\theta$ at each time-step $t$, given by:

$$\mathcal{L} = \underset{t, \mathbf{x}_0, \epsilon}{\mathbb{E}} \underbrace{\left[\lambda_t \|\mathbf{x}_0 - \left(u_\theta(\sqrt{\bar{\alpha}_t}\mathbf{x}_0 + \sqrt{1 - \bar{\alpha}_t}\epsilon, t) + w_t g(\bar{\mathbf{z}}_t, t)\right)\|^2\right]}_{\mathcal{L}_t}, \quad (5)$$

where $\lambda_t, w_t$ are time-step weight and compensate strength. We follow PDAE Zhang et al. (2022) to use fixed values computed from the variance schedule $\{\beta_t\}_{t=1}^T$ (details in Appendix).

To see the effect of Eq. 5, consider $t = 1$, only $\mathbf{z}_1$ is used to compensate for reconstruction error, hence $\mathbf{z}_1$ effectively captures a (fine-grained) modular attribute $\mathcal{Z}_i$ with $t(\mathcal{Z}_i) = 1$ (lost at $t = 1$) (such $\mathcal{Z}_i$ always exists for some threshold $\tau$ in our definition). At $t = 2$, the cumulatively lost attributes correspond to $\mathcal{Z}_i$ and $\mathcal{Z}_j$, where $t(\mathcal{Z}_j) = 2$. As $\mathbf{z}_1$ already captures the attribute $\mathcal{Z}_i$,

$\mathbf{z}_2$ is encouraged to further learn the attribute $\mathcal{Z}_j$ to minimize the reconstruction error at $t = 2$. Eventually with a large enough $T$, $\mathbf{x}_T$ becomes pure noise and all attributes are lost. By the above induction, $\{\mathbf{z}_i\}_{i=1}^T$ captures all the hidden attributes as a modular feature. In contrast, the most related work PDAE differs from Eq. 5 by replacing $g(\bar{\mathbf{z}}_t)$ with $g(\bar{\mathbf{z}})$. This means that PDAE uses the full-dimensional feature $\mathbf{z}$ to compensate the reconstruction error at all time-steps, which fails to leverage the inductive bias discussed in Section 4.1. We show in Section 5 that this simple change leads to drastic differences in both inference and generation tasks.

**Implementation Details**. We highlight some design considerations with ablations in Section 5.2.

- Number of Subsets $k$. In practice, the number of subsets for partitioning $\mathbf{z}$ can be smaller than $T$, *e.g.*, we use feature dimension $d = 512$ that is smaller than $T = 1,000$. This means that adjacent time-steps may share a modular feature $\mathbf{z}_i$. We compare the choice of $k$ later.
- Partition Strategy. The simplest way is to use the same number of feature dimensions for all modular features $\mathbf{z}_i$ (*i.e.*, balanced). However, we empirically find that features corresponding to time-step 100-300 requires more dimensions to improve convergence of Eq. 5. We design an imbalanced partition strategy and compare it with the balanced one.
- Optimization Strategy. In training, we experiment detaching the gradient of $\mathbf{z}_1, \ldots, \mathbf{z}_{t-1}$ to form $\bar{\mathbf{z}}_t$, *i.e.*, training only $\mathbf{z}_t$ at time-step $t$. We find that it leads to improved disentanglement quality at the cost of slower convergence.

## 5 EXPERIMENTS

### 5.1 SETTINGS

**Datasets**. We choose real-world datasets to validate if DiTi learns a disentangled representation of the generative attributes: 1) Celebrity Faces Attributes (CelebA) Liu et al. (2015) is a large-scale face attributes dataset. Each face image has 40 binary attribute labels, delineating facial features and characteristics, such as expressions, accessories, and lighting conditions. 2) Flickr-Faces-HQ (FFHQ) Karras et al. (2019) contains 70,000 high-quality face images obtained from Flickr. 3) We additionally used the Labeled Faces in the Wild (LFW) dataset Huang et al. (2007) that provides continuous attribute labels. 4) Bedroom is part of the Large-scale Scene UNderstanding (LSUN) dataset Yu et al. (2015) that contains around 3 million images. We apply off-the-shelf image classifiers following Yang et al. (2021) to determine scene attribute labels.

**Evaluation Protocols**. Our evaluation is based on the properties of a disentangled representation (Section 3.1). For **inference**, we perform unsupervised learning on CelebA train split or FFHQ and test the linear-probe attribute prediction accuracy on CelebA test split, measured by Average Precision (AP). We also evaluate the challenging attribute regression task using LFW, where the metrics are Pearson correlation coefficient (Pearson's r) and Mean Squared Error (MSE). For **counterfactual generation**, we extend the conventional interpolation and manipulation technique Preechakul et al. (2022) to tailor for our needs: 1) Instead of interpolating the whole features of an image pair, we interpolate only a feature subset while keeping its complement fixed. With a disentangled representation, only the attribute captured by the subset will change in the generated counterfactuals, *e.g.*, changing only expression between two faces. 2) In manipulation, an image is edited by altering its feature along the direction of an attribute classifier weight, where the classifier is trained on whole features. We constrain the classifier by ProbMask Zhou et al. (2021) to use a small subset of the most discriminative feature dimensions (32 out of 512 dimensions). For a disentangled representation, this should have minimal impact on the generation quality, as each single attribute is encoded in a compact feature subset. Detailed algorithms are in Appendix.

**Baselines** include 3 representative lineups: 1) Conventional unsupervised disentanglement methods $\beta$-TCVAE Chen et al. (2018) and IB-GAN Jeon et al. (2021); 2) Self-supervised learning methods SimCLR Chen et al. (2020); 3) The most related works Diff-AE Preechakul et al. (2022) and PDAE Zhang et al. (2022) that also learn a representation by making up DM's reconstruction errors.

**Implementation Details**. We followed the network design of encoder $f$ and decoder $g$ in PDAE and adopted its hyper-parameter settings (*e.g.*, $\lambda_t, w_t$ in Eq. 4, details in Appendix). This ensures that any emerged property of disentangled representation is solely from our leverage of the inductive bias in Section 4.1. We also used the same training iterations as PDAE, *i.e.*, 290k iterations on CelebA,

| Method | CelebA | | | FFHQ | | |
|---|---|---|---|---|---|---|
| | AP ↑ | Pearson's r ↑ | MSE ↓ | AP ↑ | Pearson's r ↑ | MSE ↓ |
| $\beta$-TCVAE Chen et al. (2018) | 45.0 | 0.378 | 0.573 | 43.2 | 0.335 | 0.608 |
| IB-GAN Jeon et al. (2021) | 44.2 | 0.307 | 0.597 | 42.8 | 0.260 | 0.644 |
| Diff-AE Preechakul et al. (2022) | 60.3 | 0.598 | 0.421 | 60.5 | 0.606 | 0.410 |
| PDAE Zhang et al. (2022) | 60.2 | 0.596 | 0.410 | 59.7 | 0.603 | 0.416 |
| SimCLR Chen et al. (2020) | 59.7 | 0.474 | 0.603 | 60.8 | 0.481 | 0.638 |
| **DiTi (Ours)** | **62.3** | **0.617** | **0.392** | **61.4** | **0.622** | **0.384** |

Table 1: AP (%) on CelebA attribute classification and Pearson's r, MSE on LFW attribute regression. The first column shows the training dataset.

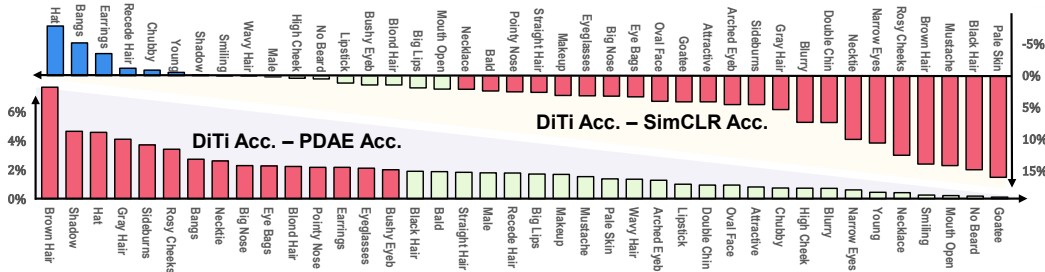

Figure 4: Improvements in attribute classification precision of our DiTi over PDAE (bottom) and over SimCLR (top). Improvements more than 2% are highlighted with red bars. Negative values are marked with blue bars.

500k iterations on FFHQ, and 540k iterations on Bedroom. Hence our DiTi training is as efficient as PDAE. Our experiments were performed on 4 NVIDIA A100 GPUs.

## 5.2 INFERENCE

**Comparison with Diff-AE and PDAE**. In Table 1, our proposed DiTi significantly outperforms the two related works. We break down the improved accuracy over PDAE on each attribute in Figure 4 bottom, where DiTi is consistently better on all attributes. The result, together with the previous analysis of Figure 2b, validates that DiTi exhibits the sample inference property of a disentangled representation. It also supports our analysis on the baselines' deficiency in Section 2, *i.e.*, they may digress to use spurious correlation to supplement reconstruction error. For example, the most improved attribute is "Brown Hair", which strongly correlates with the female gender.

**Comparison with SimCLR**. SimCLR does not perform well in attribute inference, surpassed even by PDAE. This is because SimCLR learns a representation invariant to augmentations (*e.g.*, color jittering), which disregards its related attributes, such as hair or skin color. This is verified by the large improvements of DiTi over SimCLR on "Pale Skin" accuracy in Figure 4 top. The deficiency cannot be addressed by simply removing augmentations, which leads to severe performance degradation (see Appendix). Note that SimCLR slightly outperforms DiTi on some unnoticeable attributes (*e.g.*, earrings). This suggests that some fine-grained attributes are not well disentangled. To tackle this, we will improve model expressiveness (*e.g.*, using Stable Diffusion Rombach et al. (2022) as the frozen DM) and explore more advanced training objectives Song et al. (2023) as future work.

**Comparison with $\beta$-TCVAE and IB-GAN**. It is perhaps not surprising that existing unsupervised disentanglement methods perform poorly. Their previous successes are limited to synthetic datasets, and their models are far less expressive compared to DM. In fact, their pursuit of a disentangled representation by information bottleneck proves to be ever-elusive Locatello et al. (2019).

| Imbalance | Detach | $k$ | AP ↑ | Pearson's r ↑ | MSE ↓ |
|---|---|---|---|---|---|
| ✓ | | 16 | 62.1 | 0.619 | 0.389 |
| ✓ | | 32 | 62.0 | 0.615 | 0.392 |
| ✓ | | 64 | 62.3 | 0.617 | 0.392 |
| ✓ | | 128 | 61.9 | 0.616 | 0.391 |
| | | 64 | 61.9 | 0.604 | 0.410 |
| | ✓ | 64 | 62.5 | 0.590 | 0.422 |
| ✓ | ✓ | 64 | 62.4 | 0.604 | 0.405 |

Table 2: Ablations on DiTi designs on CelebA. *Imbalance* and *Detach*: using our partition and optimization strategy.

**Ablation on #Subsets** $k$. From Table 2 first 4 lines, we observe that inference results are not sensitive to the subset number $k$ (details in Section 4.2). We choose $k = 64$ with the highest AP.

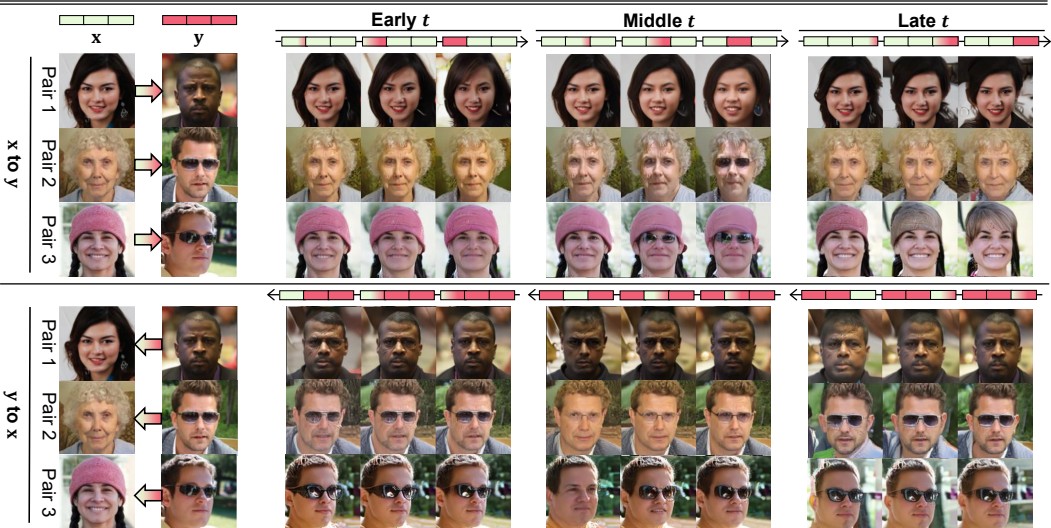

Figure 5: Counterfactual generations on FFHQ by interpolating a feature subset of 3 image pairs $(\mathbf{x}, \mathbf{y})$. We partition $\{\mathbf{z}_i\}_{i=1}^{T}$ according to early $t \in (0, T/3]$, middle $t \in (T/3, 2T/3]$ and late $t \in (2T/3, T]$. For each subset, we show results in 2 directions $(\mathbf{x} \to \mathbf{y}, \mathbf{y} \to \mathbf{x})$ under 3 interpolation scales (color gradient↑, scale↑).

This means that our model can learn up to 64 hidden attributes, whose combinations are certainly diverse enough for our chosen datasets.

**Ablation on Partition & Optimization Strategy**. We devise an imbalanced partition strategy to allocate more feature dimensions for $\mathbf{z}_i$ corresponding to $t \in \{100, \ldots, 300\}$, as this time-step range contributes the most to the overall loss (Figure A9). This leads to faster convergence and improved performance in Table 2 (line 3 vs. line 5). For optimization strategy, we tried detaching the gradient of $\mathbf{z}_1, \ldots, \mathbf{z}_{t-1}$ at time-step $t$ to prevent them from capturing the lost attribute at $t$. Yet it leads to slower convergence, which especially hurts the challenging regression task. We leave it as future work to explore more principled ways of improving feature modularity.

## 5.3 COUNTERFACTUAL GENERATION

**FFHQ Results**. Figure 5 shows DiTi generations by interpolating a feature subset while fixing its complement. Different from full-feature interpolation (Figure 7), there are two directions: $\mathbf{x} \to \mathbf{y}$ fixes the complement to the corresponding feature values of $\mathbf{x}$, and $\mathbf{y} \to \mathbf{x}$ fixes it to those of $\mathbf{y}$. By identifying how images are changing during interpolation (left to right for top, right to left for bottom) and comparing the changes across the 3 columns, we have several observations:

1. Interpolating different subsets modifies different attributes. For example, subset of early $t$ mainly controls micro-expressions, such as mouth corners, cheeks, or eye details. The interpolation hardly changes a person's identity, but certainly moves one's temperament towards the other person. In contrast, subset of middle $t$ is more responsible for identity-related attributes (*e.g.*, proportion of facial features) and accessories (*e.g.*, eyeglasses). Finally, late $t$ controls attributes such as hairstyle, pose, and even position of the face in the image (*e.g.*, the bottom right face moves towards center).

2. From early to late $t$, the corresponding feature subset controls a more coarse-grained feature, where the granularity is defined based on the pixel-level changes. For example, changing hairstyle or removing hat (subset of late $t$) modifies more pixels than changing facial feature proportions (subset of middle $t$). This is in line with our theoretical analysis in Section 4.1. Notably, both our feature subset partition and the correspondence between subsets and attributes granularity are known a priori, without relying on manual discovery techniques such as latent traversal.

3. Our generated counterfactuals have minimal distortion or artifact, and demonstrate modular properties (*e.g.*, subset of late $t$ controls pose with minimal impact on facial features and accessories). These are strong proofs that DiTi learns a disentangled representation Besserve et al. (2020). We show in Appendix that all generative baselines do not exhibit these traits.

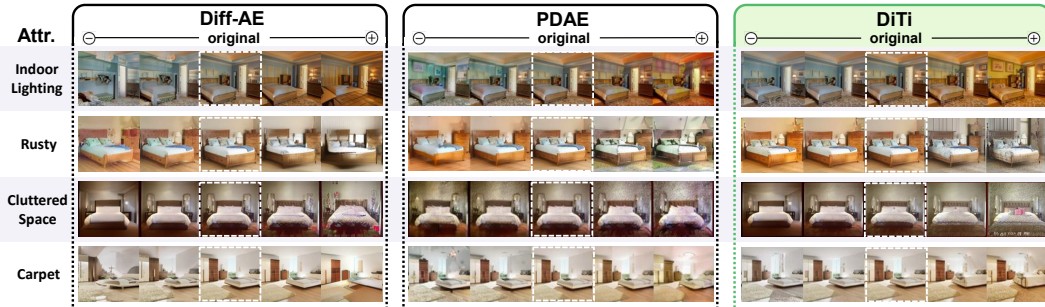

Figure 6: Counterfactual generations on Bedroom by manipulating 32 out of 512 feature dimensions.

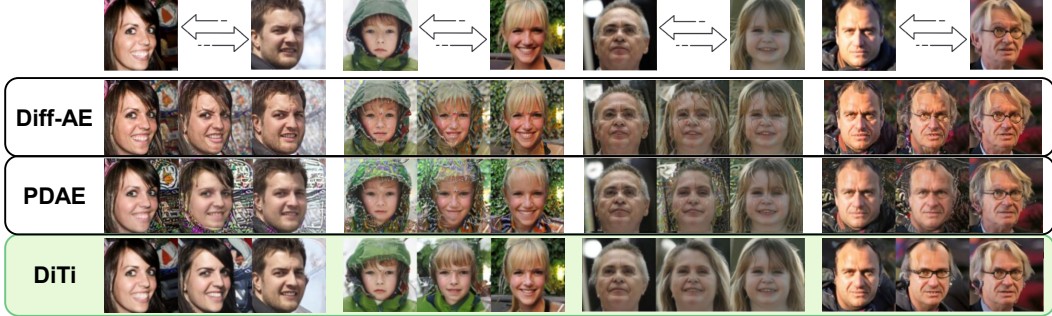

Figure 7: Results of interpolating the whole feature on FFHQ. Baselines have distortions during interpolation.

**Bedroom Manipulation**. Bedroom is a more challenging dataset, where room scenes are much more complex than facial images (*e.g.*, various layouts, decorations, objects). As shown in Figure 6, DiTi is the only method that generates faithful counterfactuals. 1st row: only DiTi can alter lighting without making significant changes to room furniture. 2nd row: baselines fail to modify the rustic feeling without artifacts or changing room layouts. 3rd row: DiTi makes the room more cluttered by introducing additional objects. Last row: only DiTi adds an additional carpet.

**Whole-feature Interpolation**. Under this conventional setting, as shown in Figure 7, only DiTi smoothly interpolates between the attributes of each image pair, with minimal distortion and artifact along the trajectories. This means that our DiTi can generate faithful counterfactuals (*i.e.*, valid images in the support of data distribution), which validates our disentanglement quality Besserve et al. (2020). For example, if a feature only captures the attribute "smile" (*i.e.*, disentangled), interpolating this feature between a smiling person and a non-smiling one leads to the gradual transition of expression. However, if a feature entangles multiple attributes, such interpolation will alter all of them or even cause distortion when the interpolated feature is out of the data distribution.

## 6 CONCLUSION

We presented a novel unsupervised method to learn a disentangled representation, which leverages the inductive bias of diffusion time-steps. In particular, we reveal an inherent connection between time-step and hidden modular attributes that generate data faithfully, enabling a simple and effective approach to disentangle the attributes by learning a time-step-specific feature. The learned feature improves downstream inference and enables counterfactual generation, validating its disentanglement quality. As future work, we will seek additional inductive bias to improve disentanglement, *e.g.*, using text as a disentangled template by exploring text-to-image diffusion models, and devise practical optimization techniques to enable faster convergence.

## 7 ACKNOWLEDGEMENT

This research is supported by Microsoft Research Asia, the National Research Foundation, Singapore under its AI Singapore Programme (AISG Award No: AISG2-RP-2021-022), MOE AcRF Tier 2 (MOE2019-T2-2-062), Wallenberg-NTU Presidential Postdoctoral Fellowship, the Lee Kong Chian (LKC) Fellowship fund awarded by Singapore Management University, and the DSO Research Grant (Fund Code MG22C03).

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
