# Appendix

This is the Appendix for "Exploring Diffusion Time-steps for Unsupervised Representation Learning". Table .1 summarizes the abbreviations and the symbols used in the main paper.

This appendix is organized as follows:

- Section A discusses the limitation and broader impact of our work.
- Section B provides additional details on the DM formulation and the hyper-parameter choice in Eq. (6).
- Section C gives the full proof to our Theorem, and shows the sufficiency of disentangled representation in minimizing Eq. (6).
- Section D presents the algorithm for training, counterfactual generation and manipulation, the network architecture and additional training details.
- Section E gives the standard deviation of the results in Table 1, manipulation and counterfactual generation results on Bedroom, additional results on FFHQ and CelebA, and interpolation & manipulation results by modifying the full-dimensional feature $\mathbf{z}$.

| Abbreviation/Symbol | Meaning |
|---|---|
| *Abbreviation* | |
| DM | Denoising Diffusion Probabilistic Model |
| OVL | Overlapping Coefficient |
| AP | Average Precision |
| MSE | Mean Squared Error |
| slerp | Spherical linear interpolation |
| lerp | Linear interpolation |
| *Symbol in Theory* | |
| $\mathcal{X}$ | Sample space |
| $\mathcal{Z}$ | Vector space |
| $\mathcal{Z}_i$ | Modular attribute |
| $g_i$ | Group element acting on $\mathcal{Z}_i$ |
| $\Phi$ | Injective mapping $\mathcal{Z} \to \mathcal{X}$ |
| $f$ | Disentangled representation $\mathcal{X} \to \mathcal{Z}$ |
| erf | Error function |
| *Symbol in Algorithm* | |
| $\mathbf{x}_0$ | Original sample |
| $\mathbf{x}_t$ | Noisy sample after $t$ forward step |
| $q(\cdot)$ | Distribution in the encoding process |
| $p_\theta(\cdot)$ | Distribution in the $\theta$-parameterized decoding process |
| $\theta$ | Parameter of U-Net |
| $u_\theta$ | $\theta$-parameterized U-Net |
| $\hat{\mathbf{x}}_0$ | Reconstructed $\mathbf{x}_0$ |
| $\mathbf{z}_i$ | $i$-th modular attribute value |
| $\bar{\mathbf{z}}_i$ | $[\mathbf{z}_1, \ldots, \mathbf{z}_i, 0, \ldots, 0]$ |
| $T$ | Total time-steps |
| $\beta_1, \ldots, \beta_T$ | Variance schedule |
| $\alpha_t$ | $1 - \beta_t$ |
| $\bar{\alpha}_t$ | $\prod_{s=1}^{t} \alpha_s$ |

Table .1: List of abbreviations and symbols used in the paper.

## A  Limitation and Broader Impact

**Limitation**. A disentangled representation is a sufficient condition to minimize our objective in Eq. (6), but not a necessary one. In particular, let $t(\mathcal{Z}_i) < t(\mathcal{Z}_j)$. At $t = t(\mathcal{Z}_i)$, attribute $\mathcal{Z}_i$ is lost with a larger degree compared to $\mathcal{Z}_j$. Hence while Eq. (6) trains $\mathbf{z}_t$ to mainly capture $\mathcal{Z}_i$, it may not fully remove $\mathcal{Z}_j$. Nevertheless, the purpose of this study is to highlight the inductive bias by diffusion time-step and explore its potential, and we hope to address this limitation as future work.

**Broader Impact**. While our learned model (encoder and decoder) can be used to generate synthetic data for malicious purposes, researchers have built models to predict fake content accurately. Moreover, the focus of our study is not improving the generation fidelity, but to learn disentangled representation, which leads to robust and fair AI models resilient to spurious correlations.

## B  Additional Details for Approach

**Closed Form of** $q\left(\mathbf{x}_{t-1}|\mathbf{x}_t, u_\theta(\mathbf{x}_t, t)\right)$. Given by $\mathcal{N}(\mathbf{x}_{t-1}|\tilde{\boldsymbol{\mu}}_t(\mathbf{x}_t, \mathbf{x}_0), \tilde{\beta}_t \mathbf{I})$, where

$$\tilde{\boldsymbol{\mu}}_t(\mathbf{x}_t, \mathbf{x}_0) = \frac{\sqrt{\bar{\alpha}_{t-1}}\beta_t}{1 - \bar{\alpha}_t}\mathbf{x}_0 + \frac{\sqrt{\alpha_t}(1 - \bar{\alpha}_{t-1})}{1 - \bar{\alpha}_t}\mathbf{x}_t, \quad \tilde{\beta}_t = \frac{1 - \bar{\alpha}_{t-1}}{1 - \bar{\alpha}_t}\beta_t. \tag{B.1}$$

**Equivalent Formulation**. The simplified objective in DDPM [3] is given by:

$$\mathcal{L}_{DM} = \mathop{\mathbb{E}}_{t,\mathbf{x}_0,\epsilon} \|\epsilon - \epsilon_\theta(\sqrt{\bar{\alpha}_t}\mathbf{x}_0 + \sqrt{1 - \bar{\alpha}_t}\epsilon, t)\|^2, \tag{B.2}$$

where $\epsilon_\theta$ is a $\theta$-parameterized U-Net [6] to predict the added noise. From Eq. (2), we have

$$\epsilon = \frac{\mathbf{x}_t - \sqrt{\bar{\alpha}_t}\mathbf{x}_0}{\sqrt{1 - \bar{\alpha}_t}}, \quad \epsilon_\theta = \frac{\mathbf{x}_t - \sqrt{\bar{\alpha}_t}u_\theta}{\sqrt{1 - \bar{\alpha}_t}}, \tag{B.3}$$

where we slightly abuse the notation to denote the reconstructed $\mathbf{x}_0$ from our U-Net as $u_\theta$. Taking Eq. (B.3) into Eq. (B.2) yields Eq. (3).

**Time-step Weight** $\lambda_t$ **and Compensate Strength** $w_t$. The PDAE objective is given by

$$\mathcal{L}_{PDAE} = \mathop{\mathbb{E}}_{t,\mathbf{x}_0,\epsilon} \left[\lambda_t^p \|\epsilon - \epsilon_\theta(\sqrt{\bar{\alpha}_t}\mathbf{x}_0 + \sqrt{1 - \bar{\alpha}_t}\epsilon, t) + w_t^p g(\mathbf{z}, t)\|^2\right], \tag{B.4}$$

where

$$\lambda_t^p = \left(\frac{1}{1 + \text{SNR}(t)}\right)^{0.9} \left(\frac{\text{SNR}(t)}{1 + \text{SNR}(t)}\right)^{0.1}, \text{SNR}(t) = \frac{\bar{\alpha}_t}{1 - \bar{\alpha}_t}, w_t^p = \frac{\sqrt{\alpha_t}(1 - \bar{\alpha}_{t-1})}{\sqrt{1 - \bar{\alpha}_t}}. \tag{B.5}$$

Taking Eq. (B.3) into Eq. (B.4) yields:

$$\lambda_t = \frac{\bar{\alpha}_t}{1 - \bar{\alpha}_t}\lambda_t^p, \ w_t = \frac{\sqrt{1 - \bar{\alpha}_t}}{\sqrt{\bar{\alpha}_t}}w_t^p = \sqrt{\frac{\alpha_t}{\bar{\alpha}_t}}(1 - \bar{\alpha}_t). \tag{B.6}$$

## C  Theory

**Full Proof to the Theorem**. We list the theorem below for reference.

**Theorem**. (Attribute Loss and Time-step) *1) For each $\mathcal{Z}_i$, there exists a smallest time-step $t(\mathcal{Z}_i)$, such that $\mathcal{Z}_i$ is lost with degree $\tau$ at each $t \in \{t(\mathcal{Z}_i), \ldots, T\}$. 2) $\exists\{\beta_i\}_{i=1}^T$ such that $t(\mathcal{Z}_i) > t(\mathcal{Z}_j)$ whenever $\|\mathbf{x}_0 - g_i \cdot \mathbf{x}_0\|$ is first-order stochastic dominant over $\|\mathbf{x}_0 - g_j \cdot \mathbf{x}_0\|$ with $\mathbf{x}_0 \sim \mathcal{X}$ uniformly.*

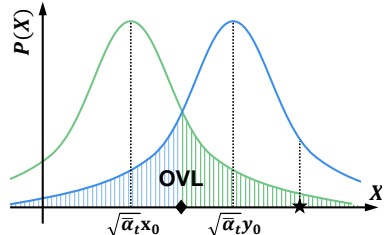

Figure C.1: The PDFs of $q(\mathbf{x}_t|\mathbf{x}_0)$ (green) and $q(\mathbf{y}_t|\mathbf{y}_0)$ (blue). Without loss of generality, we consider the 1-D case of $y_0 > x_0$. Their means are computed from Eq. (2).

*Proof.* We start by showing $\text{Err}(\mathbf{x}_0, \mathbf{y}_0, t) = \frac{1}{2}\text{OVL}\left(q(\mathbf{x}_t|\mathbf{x}_0), q(\mathbf{y}_t|\mathbf{y}_0)\right)$. Without loss of generality, we show a 1-D sample space $\mathcal{X}$ in Figure C.1. The minimum $\text{Err}_\theta$ is obtained when given each noisy sample $\mathbf{x}$,

DM reconstructs towards $\mathbf{x}_0$ if $q(\mathbf{x}|\mathbf{x}_0) > q(\mathbf{x}|\mathbf{y}_0)$ and vice versa for $\mathbf{y}_0$, *e.g.*, reconstructing $\star$ as $\mathbf{y}_0$. However, this maximum likelihood estimation fails when a noisy sample is drawn from $q(\mathbf{x}_t|\mathbf{x}_0)$ (green PDF), but with a value larger than the intersection point of the two PDFs ($\blacklozenge$), and similar arguments go for $q(\mathbf{y}_t|\mathbf{y}_0)$ (blue PDF). The error rate caused by the two failure cases corresponds to the green shaded area and blue one, respectively, leading to an average $\text{Err}_\theta$ of $\frac{1}{2}$ of the OVL.

To compute the OVL, it is trivial in the 1-D case by leveraging the Cumulative Distribution Function (CDF) of Gaussian distribution. Given that the two distributions have equal variance from Eq. (2), the intersection point is given by $\frac{\sqrt{\bar{\alpha}_t}\mathbf{x}_0 + \sqrt{\bar{\alpha}_t}\mathbf{y}_0}{2}$. For a Gaussian distribution $\mathcal{N}(\mu, \sigma^2)$, its CDF is given by $\frac{1}{2}\left[1 + \text{erf}(\frac{x-\mu}{\sqrt{2}\sigma})\right]$. Combining two results, one can easily show that the blue shaded area, corresponding to half of the OVL, or $\text{Err}(x_0, y_0, t)$, is given by:

$$\text{Err}(x_0, y_0, t) = \frac{1}{2}\text{OVL}\left(q(x_t|x_0), q(y_t|y_0)\right) = \frac{1}{2}\left[1 - \text{erf}\left(\frac{\sqrt{\bar{\alpha}_t}(y_0 - x_0)}{2\sqrt{2(1-\bar{\alpha}_t)}}\right)\right]. \quad \text{(C.1)}$$

To generalize the results to multi-variate Gaussian distributions, we use the results in [5], which shows that by projecting the data to Fisher's linear discriminant axis, the OVL defined on the discriminant densities is equal to that defined on the multivariate densities. Specifically, the mean of the discriminant densities are given by

$$\mu_0 = \sqrt{\bar{\alpha}_t}(\mathbf{y}_0 - \mathbf{x}_0)^\top \Sigma^{-1}\mathbf{x}_0, \; \mu_1 = \sqrt{\bar{\alpha}_t}(\mathbf{y}_0 - \mathbf{x}_0)^\top \Sigma^{-1}\mathbf{y}_0, \quad \text{(C.2)}$$

where $\Sigma = \beta_t\mathbf{I}$. The common variance of the discriminant densities is given by $\sqrt{\bar{\alpha}_t}(\mathbf{y}_0 - \mathbf{x}_0)^\top \Sigma^{-1}(\mathbf{y}_0 - \mathbf{x}_0)$. Following the calculation steps to compute OVL for the 1-D case, one can show that for both 1-D and multi-variate case, we have

$$\text{Err}(\mathbf{x}_0, \mathbf{y}_0, t) = \frac{1}{2}\text{OVL}\left(q(\mathbf{x}_t|\mathbf{x}_0), q(\mathbf{y}_t|\mathbf{y}_0)\right) = \frac{1}{2}\left[1 - \text{erf}\left(\frac{\|\sqrt{\bar{\alpha}_t}(\mathbf{y}_0 - \mathbf{x}_0)\|}{2\sqrt{2(1-\bar{\alpha}_t)}}\right)\right]. \quad \text{(C.3)}$$

As $\bar{\alpha}_t$ decreases with an increasing $t$ from Eq. (2), and the error function $\text{erf}(\cdot)$ is strictly increasing, $\text{Err}(\mathbf{x}_0, \mathbf{y}_0, t)$ is strictly increasing in $t$ given any $\mathbf{x}_0, \mathbf{y}_0$. Hence $\mathbb{E}_{\mathbf{x}_0 \in \mathcal{X}}\left[\text{Err}(\mathbf{x}_0, \mathbf{y}_0 = g_i \cdot \mathbf{x}_0, t)\right] \geq \mathbb{E}_{\mathbf{x}_0 \in \mathcal{X}}\left[\text{Err}(\mathbf{x}_0, \mathbf{y}_0 = g_i \cdot \mathbf{x}_0, t(\mathcal{Z}_i))\right]$ for every $t \geq t(\mathcal{Z}_i)$, which completes the proof of Theorem 1.

Given that $\text{erf}(\cdot)$ is strictly increasing and $\|\mathbf{x}_0 - g_i \cdot \mathbf{x}_0\|$ is first-order stochastic dominant over $\|\mathbf{x}_0 - g_j \cdot \mathbf{x}_0\|$, we have $\mathbb{E}_{\mathbf{x}_0 \in \mathcal{X}}\left[\text{Err}(\mathbf{x}_0, g_i \cdot \mathbf{x}_0, t)\right] > \mathbb{E}_{\mathbf{x}_0 \in \mathcal{X}}\left[\text{Err}(\mathbf{x}_0, g_j \cdot \mathbf{x}_0, t)\right]$ at every time-step $t$ using Eq. (C.3). Hence $t(\mathcal{Z}_i) > t(\mathcal{Z}_j)$ under any variance schedule $\{\beta_i\}_{i=1}^T$ such that $\mathcal{Z}_i$ is *not* lost at $t(\mathcal{Z}_j)$, completing the proof of Theorem 2.

**Disentangled Representation Minimizes Eq. (6)**. Suppose that we have a disentangled representation $f$ that maps images to $\{\mathbf{z}_i\}_{i=1}^T$. Without loss of generality, we assume an attribute order condition where $\mathbf{z}_1, \ldots, \mathbf{z}_T$ take the order such that $\{\mathbf{z}_i\}_{i=1}^t$ makes up the cumulatively lost attributes at each $t$, *i.e.*, $t(\mathcal{Z}_i) \leq t, \forall i \in \{1, \ldots, t\}$ and $t(\mathcal{Z}_i) \geq t, \forall i \in \{t, \ldots, T\}$. Hence given $t$, for each $g_i$ such that $\mathbb{E}_{\mathbf{x}_0 \in \mathcal{X}}\left[\text{Err}(\mathbf{x}_0, \mathbf{y}_0 = g_i \cdot \mathbf{x}_0, t)\right] \geq \tau$, we have $[f(\mathbf{x}_0)]_t \neq [f(g_i \cdot \mathbf{x}_0)]_t, \forall \mathbf{x}_0 \in \mathcal{X}$, where $[\cdot]_t$ extracts $\{\mathbf{z}_i\}_{i=1}^t$ from $\{\mathbf{z}_i\}_{i=1}^T$. Hence there exists an decoder $g$ that maps each unique $[f(\mathbf{x}_0)]_t$ to the corresponding reconstruction error $\hat{\mathbf{x}}_0 - \mathbf{x}_0$ by the pre-trained DM. For other $g_i$, the reconstruction error is bounded by $\mathbb{E}_{\mathbf{x}_0 \in \mathcal{X}}\left[\text{Err}(\mathbf{x}_0, \mathbf{y}_0 = g_i \cdot \mathbf{x}_0, t)\right] < \tau$. Hence we prove that the reconstruction error (or attribute loss) can be arbitrarily small (up to specified $\tau$) given a disentangled representation $f$ and a variance schedule that satisfies Theorem 2 (to make sure the attribute order condition holds).

# D Additional Experiment Details

**Network Architecture**. We exactly follow the encoder and decoder design in PDAE [8] and use the same pre-trained DM. Please refer to PDAE for more details.

**Imbalanced Partition Strategy**. As shown in Figure D.1, we plot the average loss $\mathcal{L}_t$ (in the most recent 5k iterations) at each time-step $t$. It is clear that time-step 100-300 contribute the most to the overall loss. Furthermore, by comparing the loss at 5k iteration and 35k iteration, we observe that the same time-step range contributes the most to the loss minimization. We conjecture that the time-step 100-300 contains rich semantic information. On the other hand, late time-steps (*e.g.*, after $t = 500$) have smaller loss value and less loss reduction, as late time-steps have very small weight $\lambda_t$ by the design of DDPM [3]. Hence accordingly, we design an imbalanced

partition strategy to allocate more feature dimensions to time-step 100-300 and less ones to time-step 500-1000. Specifically, we assign 10, 25, 327, 100, 50 dimensions to time-step range 0-50, 50-100, 100-300, 300-500, 500-1000, respectively. Note that we only tried this dimension allocation strategy as a heuristic approach, and we did not search for an optimal strategy. Future work can explore an adaptive allocation strategy.

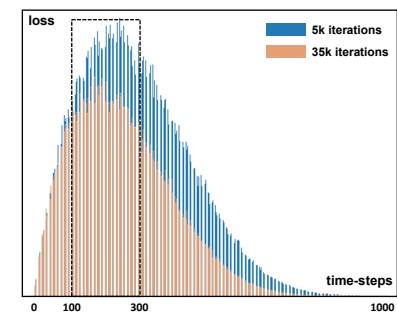

Figure D.1: Average $\mathcal{L}_t$ for each time-step $t$ in DiTi training at 5k iterations and 35k iterations.

**Optimization Strategy**. Figure D.2 compares the loss in Eq. (6) by DiTi and DiTi-Detach (*i.e.*, detaching the gradients of $\mathbf{z}_1, \ldots, \mathbf{z}_{t-1}$) throughout training. The loss reduction of DiTi-Detach is much slower as only $\frac{1}{k}$ of the feature is trained at each iteration. As shown in Table 2, this alternative optimization strategy hurts the performance when transferring the feature trained on CelebA for LFW attribute regression. We conjecture that transfer learning and regression task is more difficult, hence LFW regression is more sensitive to model convergence. However, this strategy does provide additional inductive bias towards disentanglement, as only $\mathbf{z}_t$ is trained to capture the lost attribute at time-step $t$. Hence we use DiTi for classification/regression tasks and DiTi-Detach for generation tasks. As future work, we will explore improved network design and other optimization techniques to reap the benefits of DiTi-Detach strategy without hurting convergence.

**Training Algorithm**. Please refer to Algorithm 1.

**Counterfactual Generation Algorithm**. Please refer to Algorithm 2.

**Modular Manipulation Algorithm**. In modular manipulation, we use the attribute labels to train a linear classifier that predicts a specific attribute. On CelebA [4], we use its 40 attribute labels for training. On Bedroom [7], there are no ground-truth attribute labels. We use pseudo-labels produced by an off-the-shelf attribute predictor [9] to train the attribute classifier. In particular, we adopt ProbMask [10] to constrain the classifier such that its weight has only $d'$ non-zero dimensions, where $d' < d$ (*e.g.*, $d' = 16$ or $32$ and $d = 512$). This design is to test the modularity of the feature—a specific attribute (*e.g.*, "Young") should be captured by the combination of a few modular attributes

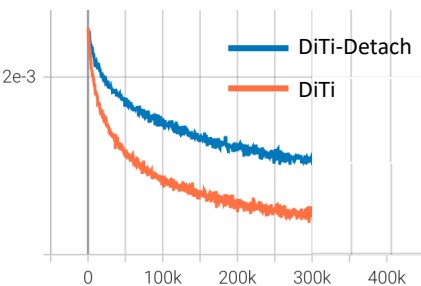

Figure D.2: Training loss of DiTi and DiTi with detach optimiaztion strategy.

$\mathbf{z}_i$, but not all. With the trained classifier for an attribute, to manipulate the attribute with scale $\lambda$ on a sample $\mathbf{x}_0$, we first obtain its feature $\mathbf{z} = f(\mathbf{x}_0)$, then push $\mathbf{z}$ along the normal vector of the decision boundary with certain scale $\lambda$, resulting in manipulated code $\mathbf{z}'$, and finally encode $\mathbf{x}_T$ back to the manipulated image by the guidance of $g(\mathbf{z}', t)$. The process is summarized in Algorithm 3.

---

**Algorithm 1:** DiTi training

---

**Input** :Training data distribution $q(\mathbf{x}_0)$, pre-trained DM $u_\theta$
**Output**:Trained encoder $f$, decoder $g$
Randomly initialize $f, g$;
**while** *not converged* **do**
    $\mathbf{x}_0 \sim q(\mathbf{x}_0)$;
    $\mathbf{z} = f(\mathbf{x}_0)$;
    Partition $\mathbf{z}$ into $\{\mathbf{z}_i\}_{i=1}^T$;
    $t \sim \text{Uniform}(1, \ldots, T)$;
    $\boldsymbol{\epsilon} \sim \mathcal{N}(\mathbf{0}, \mathbf{I})$;
    $\mathbf{x}_t = \sqrt{\bar{\alpha}_t}\mathbf{x}_0 + \sqrt{1 - \bar{\alpha}_t}\boldsymbol{\epsilon}$;
    $\bar{\mathbf{z}}_t = [\mathbf{z}_1, \ldots, \mathbf{z}_t, \mathbf{0}, \ldots, \mathbf{0}]$;
    Update $f, g$ by minimizing $\lambda_t \|\mathbf{x}_0 - (u_\theta(\mathbf{x}_t, t) + w_t g(\bar{\mathbf{z}}_t, t))\|^2$ in Eq. (6);
**return** $f, g$

**Algorithm 2:** Counterfactual generation from $\mathbf{x}_0$ to $\mathbf{x}_0'$ on subset $\mathcal{S}$ with scale $\lambda$

---

**Input** : $\mathbf{x}_0, \mathbf{x}_0'$, subset $\mathcal{S} \subset \{1, \ldots, k\}$, scale $\lambda$, pre-trained $u_\theta$, trained $f, g$, sampling sequence $\{t_i\}_{i=1}^M$ where $t_1 = 0$ and $t_M = T$
**Output**: A counterfactual image for $\mathbf{x}_0$
Compute $\mathbf{x}_T, \mathbf{x}_T'$ for $\mathbf{x}_0, \mathbf{x}_0'$ with DDIM inversion, respectively;
$\mathbf{z} = f(\mathbf{x}_0), \mathbf{z}' = f(\mathbf{x}_0')$;
Partition $\mathbf{z}$ into $\{\mathbf{z}_i\}_{i=1}^T$, $\mathbf{z}'$ into $\{\mathbf{z}_i'\}_{i=1}^T$;
$\mathbf{z}_\mathcal{S} \leftarrow \text{lerp}(\mathbf{z}, \mathbf{z}', \mathcal{S}; \lambda)$, *i.e.*, perform linear interpolation on all $\mathbf{z}_i, i \in \mathcal{S}$;
$\mathbf{x}_T \leftarrow \text{slerp}(\mathbf{x}_T, \mathbf{x}_T'; \lambda)$;
**for** $i = M, \ldots, 2$ **do**
$\quad \hat{\mathbf{x}}_0 = u_\theta(\mathbf{x}_{t_i}, t_i) + w_t g(\mathbf{z}_\mathcal{S}, t_i)$;
$\quad \mathbf{x}_{t_{i-1}} \leftarrow \sqrt{\bar{\alpha}_{t_{i-1}}} \hat{\mathbf{x}}_0 + \frac{\sqrt{1 - \bar{\alpha}_{t_{i-1}}}(\mathbf{x}_{t_i} - \sqrt{\bar{\alpha}_{t_i}} \hat{\mathbf{x}}_0)}{\sqrt{1 - \bar{\alpha}_{t_i}}}$;
**return** $\mathbf{x}_0$

---

**Algorithm 3:** Modular manipulation on $\mathbf{x}_0$ with a trained ProbMask classifier and scale $\lambda$

---

**Input** : Original $\mathbf{x}_0$, manipulation scale $\lambda$, trained ProbMask classifier with weight parameter $\mathbf{w} \in \mathbb{R}^d$, pre-trained DM $u_\theta$, trained $f, g$, standard deviation $\boldsymbol{\sigma}$ of $\mathbf{z}$ in the entire training dataset, sampling sequence $\{t_i\}_{i=1}^M$ where $t_1 = 0$ and $t_M = T$
**Output**: A counterfactual image for $\mathbf{x}_0$
Compute $\mathbf{x}_T$ for $\mathbf{x}_0$ with DDIM inversion;
$\mathbf{z} = f(\mathbf{x}_0)$;
$\mathbf{z}' = \mathbf{z} + \lambda \frac{\boldsymbol{\sigma} \cdot \mathbf{w}}{\|\mathbf{w}\|}$;
**for** $i = M, \ldots, 2$ **do**
$\quad \hat{\mathbf{x}}_0 = u_\theta(\mathbf{x}_{t_i}, t_i) + w_t g(\mathbf{z}', t_i)$;
$\quad \mathbf{x}_{t_{i-1}} \leftarrow \sqrt{\bar{\alpha}_{t_{i-1}}} \hat{\mathbf{x}}_0 + \frac{\sqrt{1 - \bar{\alpha}_{t_{i-1}}}(\mathbf{x}_{t_i} - \sqrt{\bar{\alpha}_{t_i}} \hat{\mathbf{x}}_0)}{\sqrt{1 - \bar{\alpha}_{t_i}}}$;
**return** $\mathbf{x}_0$

---

## E   Additional Experiment Results

| | Methods | AP ↑ | Pearson's r ↑ | MSE ↓ |
|---|---|---|---|---|
| **CelebA** | PDAE [8] | $60.2 \pm 0.013$ | $0.596 \pm 5.9\text{e-}4$ | $0.410 \pm 6.8\text{e-}4$ |
| | SimCLR [1] | $59.7 \pm 0.015$ | $0.474 \pm 3.7\text{e-}3$ | $0.603 \pm 3.1\text{e-}3$ |
| | SimCLR$-$Aug [1] | $34.7 \pm 0.063$ | $0.176 \pm 3.3\text{e-}3$ | $0.717 \pm 2.2\text{e-}3$ |
| | SimSiam [2] | $51.7 \pm 0.152$ | $0.464 \pm 7.2\text{e-}3$ | $0.525 \pm 4.5\text{e-}3$ |
| | **DiTi (Ours)** | $62.3 \pm 0.083$ | $0.617 \pm 4.6\text{e-}4$ | $0.392 \pm 1.0\text{e-}3$ |
| **FFHQ** | PDAE [8] | $59.7 \pm 0.030$ | $0.603 \pm 7.1\text{e-}4$ | $0.416 \pm 1.2\text{e-}3$ |
| | SimCLR [1] | $60.8 \pm 0.025$ | $0.481 \pm 4.5\text{e-}3$ | $0.638 \pm 1.9\text{e-}2$ |
| | **DiTi (Ours)** | $61.4 \pm 0.029$ | $0.622 \pm 3.9\text{e-}4$ | $0.384 \pm 2.9\text{e-}4$ |

Table E.1: AP (%) on CelebA attribute classification and Pearson's r, MSE on LFW attribute regression. Supplementary to Table 1. Standard deviations are computed on 5 independent runs.

**Standard Deviation Supplementary to Table 1**. We run the experiments in Table 1 with 5 random seeds and compute the standard deviation shown in Table E.1. Overall, the fluctuation of the results is small and the conclusions drawn from Table 1 are statistically significant.

**Attribute Manipulation on Bedroom** We show the manipulation results on Bedroom in Figure E.1 for four attributes: "indoor lighting", "rusty", "cluttered space" and "carpet". Pushing the latent code along the the normal direction is able to change the specific attribute in the image. For example, the room becomes much brighter and warm when the scale increases in the positive direction, while it becomes colder and dimmer in the negative direction. Note that here we only manipulate 32 dimensions of the latent code, much fewer than the latent code $\mathbf{z}$ has, *i.e.*, 512. It shows that our mode learns a disentangled and compact feature. In contrast, the manipulated images by PDAE often

**PDAE**                    **DiTi (Ours)**

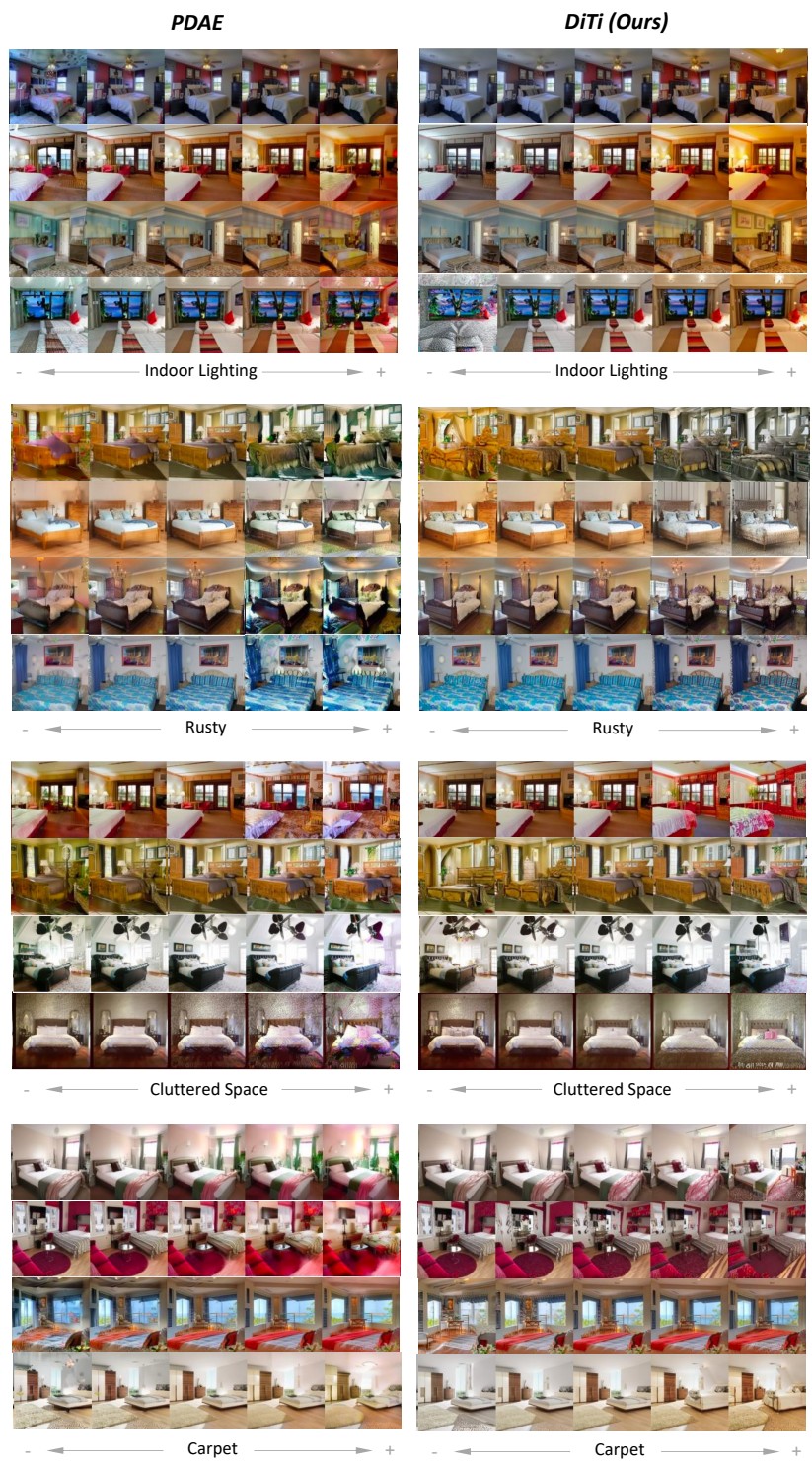

Figure E.1: Comparsions of PDAE (left column) and DiTi (right column)'s modular manipulation results on Bedroom. For both methods, we use ProbMask to constrain the classifier weight such that it has $d' = 32$ non-zero dimensions.

contain artifacts and have less meaningful edits, which further validates that PDAE entangles all the attributes.

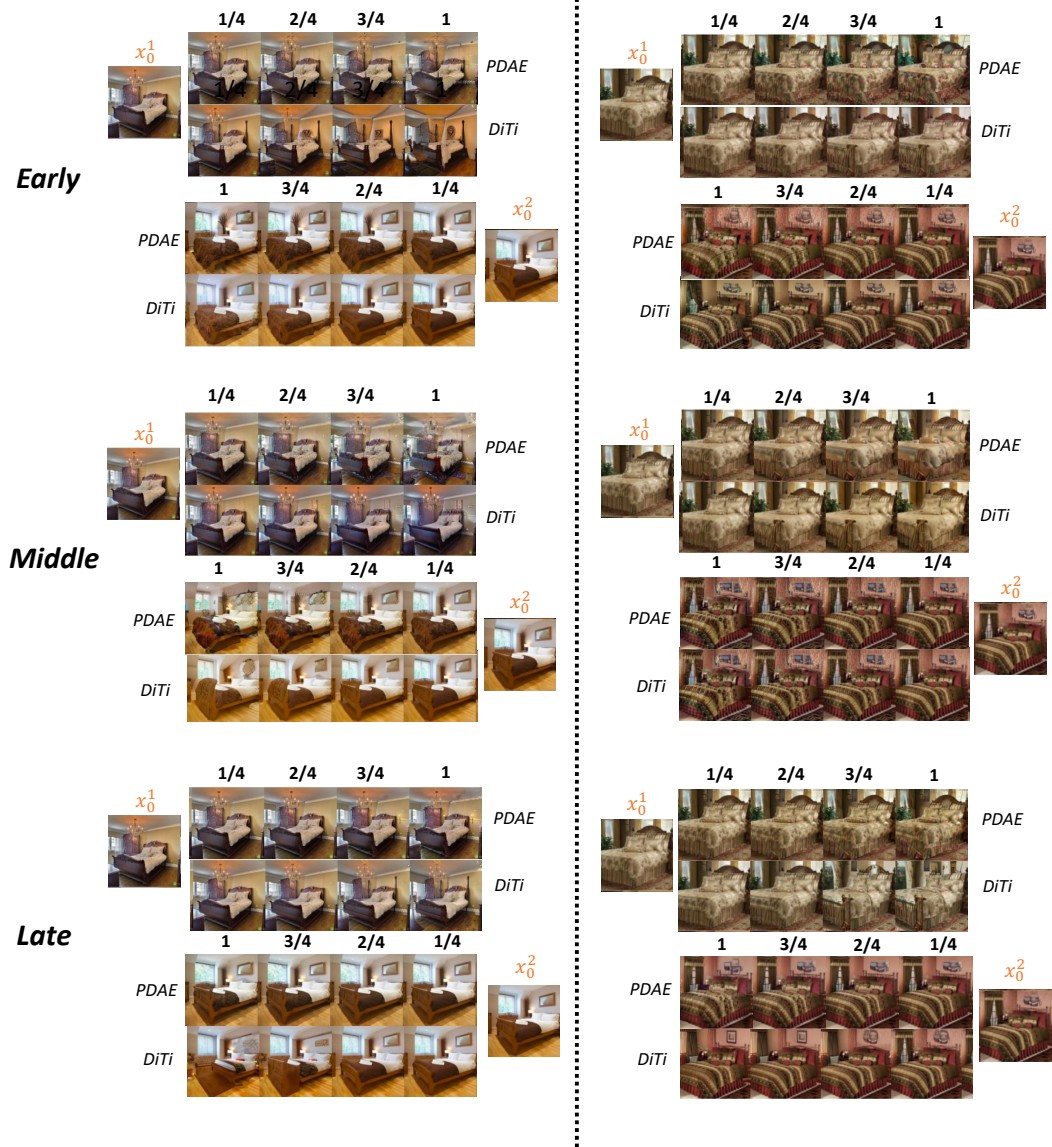

Figure E.2: Comparisions on PDAE (top row) and DiTi (bottom row)'s counterfactual generation results on Bedroom using 3 feature subsets. For each image pair in each subset, the original images, $\mathbf{x}_0^1$, $\mathbf{x}_0^2$ are placed in the top left corner and bottom right corner, and interpolated towards its counterpart in four scales (1/4, 2/4, 3/4 and 1) respectively.

**Counterfactural Generation on Bedroom** As shown in the Figure E.2, we compare the counterfactural generation results with PDAE and DiTi on three subsets. "Early", "Middle", "Late" correspond to the subset 0-8, 8-16, 24-28 respectively. We observe that each subset control some meaningful attribute in the image. Take the right column for example, in the early subset, the overall color scheme is interpolated. Interpolating the middle subset changes the shape of the end of the bed. $x_0^1$ takes more red from $x_0^2$ with the increasing of the scale. On the other hand, $x_0^2$ gets more brown color when its scale increases. Interpolating the middle subset (8-16) changes the texture of the quilt. Interpolating the late subset changes the background. $x_0^1$'s background changes from glass window to the wall that $x_0^2$ has. At the same time, $x_0^2$ changes the photo frame on the wall. Although PDAE also changes the background in the late subset, it changes the texture in the quilt at the same time.

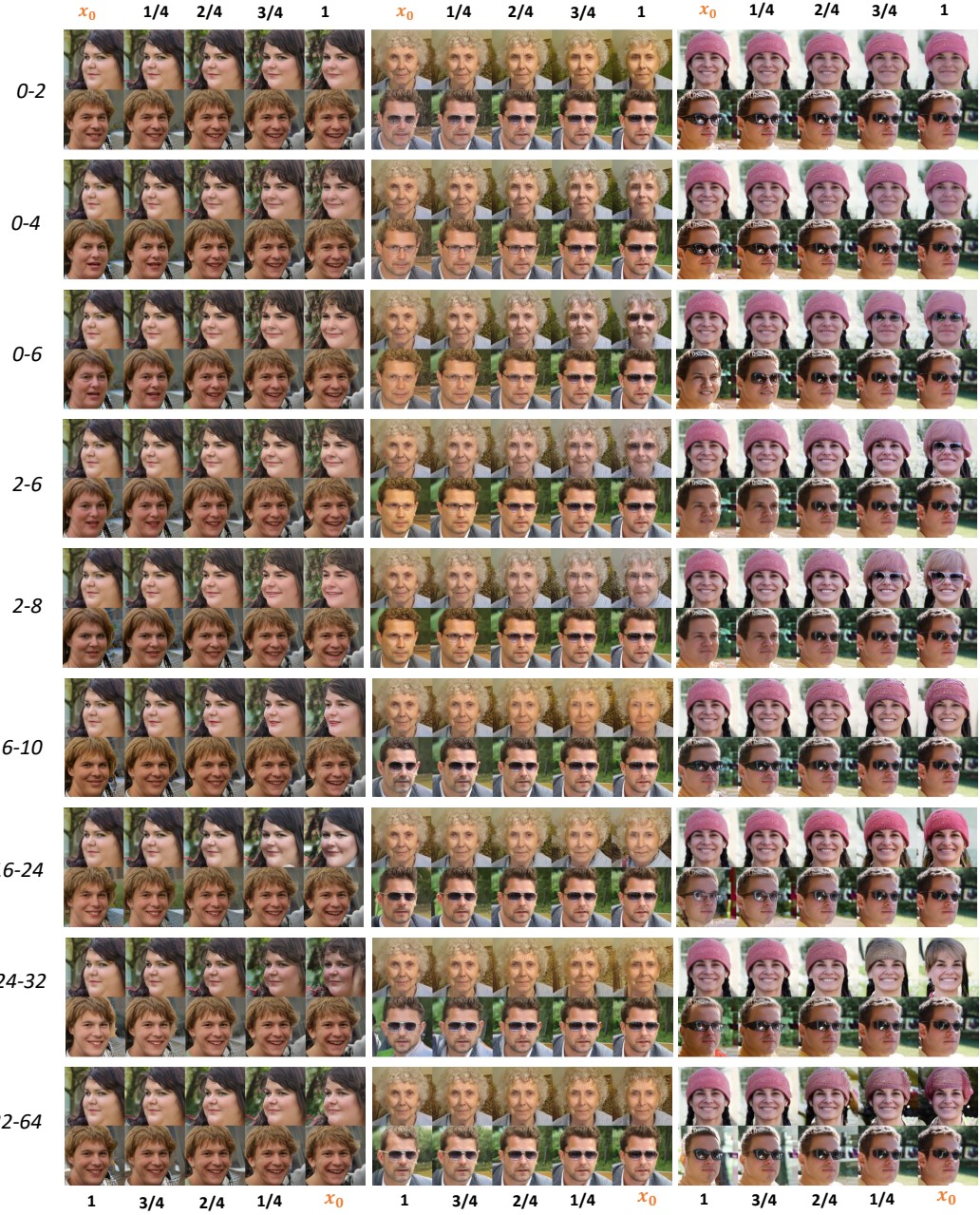

Figure E.3: DiTi's counterfactual generation results on FFHQ. For every image pair, the original images are placed on the top left and bottom right corners, and interpolated on a specific subset of $\{\mathbf{z}_i\}_{i=1}^{k}$ to gradually transition towards its counterpart with four scales (1/4, 2/4, 3/4 and 1). Each row is the result of interpolating on a subset, with the subset range displayed on the left.

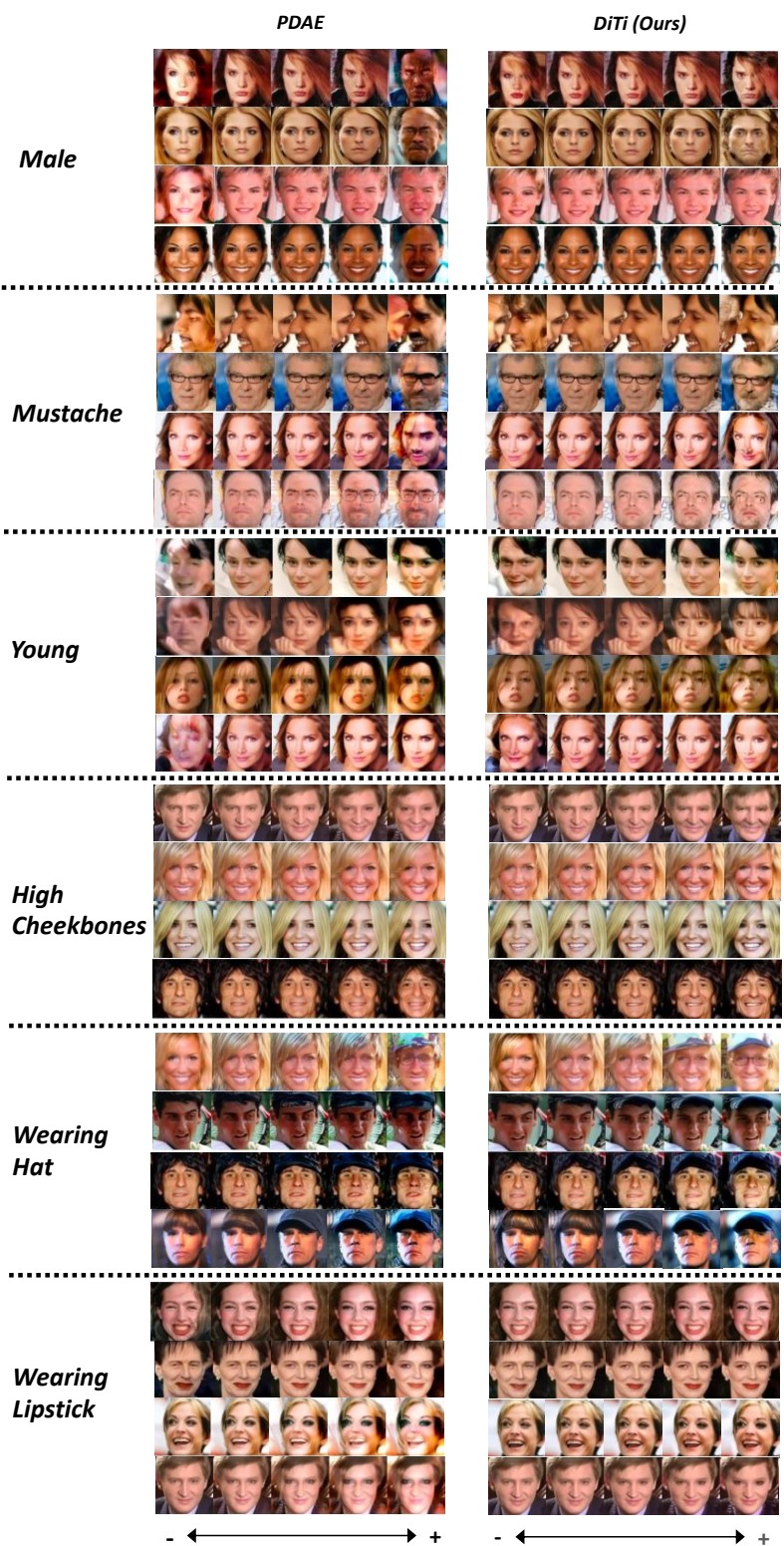

Figure E.4: Comparsions of PDAE and DiTi's modular manipulation results on CelebA. The center image in each group of 5 images corresponds to the original image $\mathbf{x}_0$. We use $d' = 128$ as the dimension of ProbMask on "Wearing Hat" attribute and $d' = 16$ for the rest.

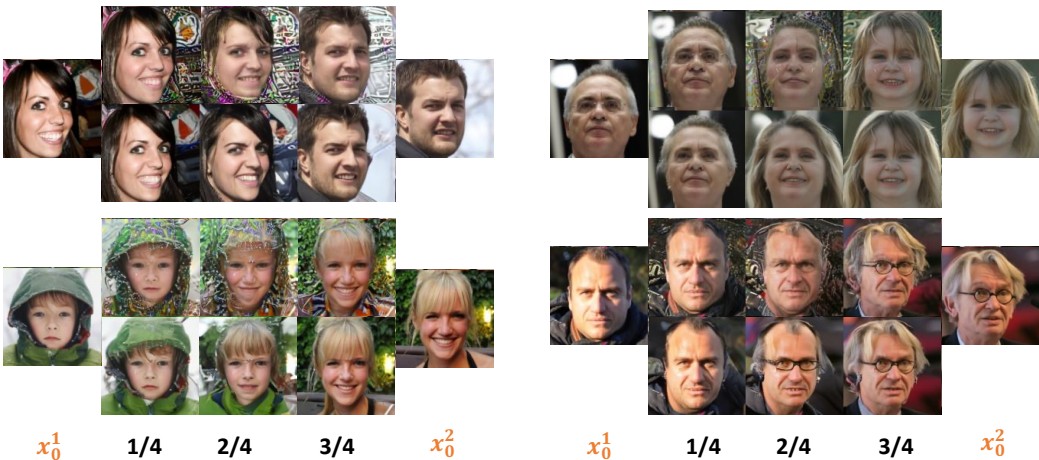

$x_0^1$    **1/4**    **2/4**    **3/4**    $x_0^2$      $x_0^1$    **1/4**    **2/4**    **3/4**    $x_0^2$

Figure E.5: Comparsions of PDAE (top row) and DiTi(bottom row)'s interpolation results on FFHQ using all dimensions of the feature **z**.

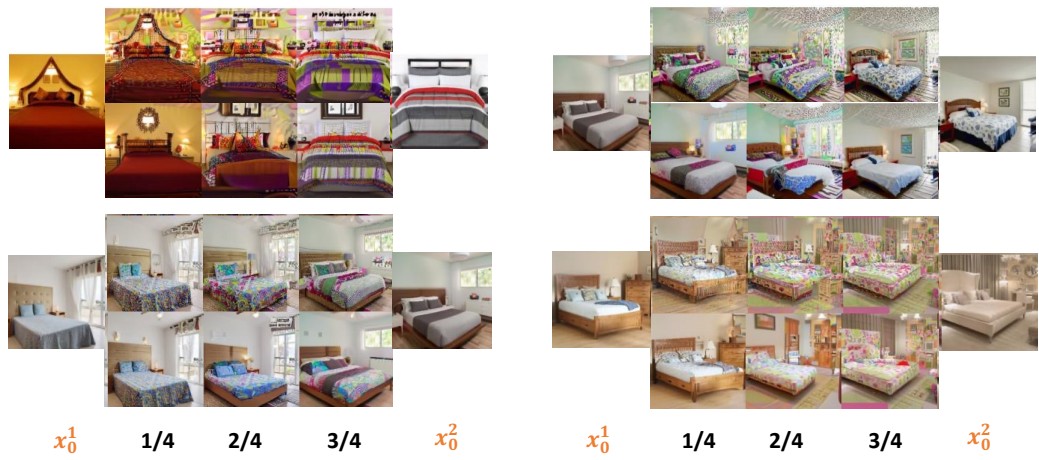

$x_0^1$    **1/4**    **2/4**    **3/4**    $x_0^2$      $x_0^1$    **1/4**    **2/4**    **3/4**    $x_0^2$

Figure E.6: Comparsions of PDAE (top row) and DiTi (bottom row)'s interpolation results on Bedroom using all dimensions of the feature **z**.

**Additional Counterfactual Generation on FFHQ**. In Figure E.3, we show additional counterfactual generations on FFHQ with more feature subsets. In particular, interpolating subset 0-2 roughly corresponds to modifying "expression" attribute (*e.g.*, center pair). Interpolating 0-4 and 0-6 additionally modifies "mouth" attribute and "eyes" attribute, respectively. This shows that the learned feature subspace $\mathcal{Z}_1, \mathcal{Z}_2, \mathcal{Z}_3$ corresponds to the three attributes, and can be combined by Cartesian product, hence verifying our disentanglement quality. Subset 6-10, 16-24, 24-32, 32-64 correspond to gradually coarser-grained attributes, "face shape", "face direction", "hairstyle", "decorations on the head", respectively, verifying our Theorem. Note from Figure D.1 that the loss is low on late feature subset due to the design of time-step weight of DDPM. Hence more feature subsets are grouped together to make more visible edits.

**Addtional Results on Modular Manipulation on CelebA**. Results are shown in Figure E.4, where our DiTi makes more meaningful edits with less artifacts compared to PDAE. In particular, we use more ProbMask dimension $d' = 128$ for attribute "Wearing Hat", as it corresponds to larger edits that involve multiple modular attributes (*e.g.*, shadows and forehead).

**Interpolation Results on FFHQ**. We run the standard interpolation experiments (*i.e.*, interpolating the entire **z** instead of a subset) with results in Figure E.5. We highlight that our DiTi outperforms

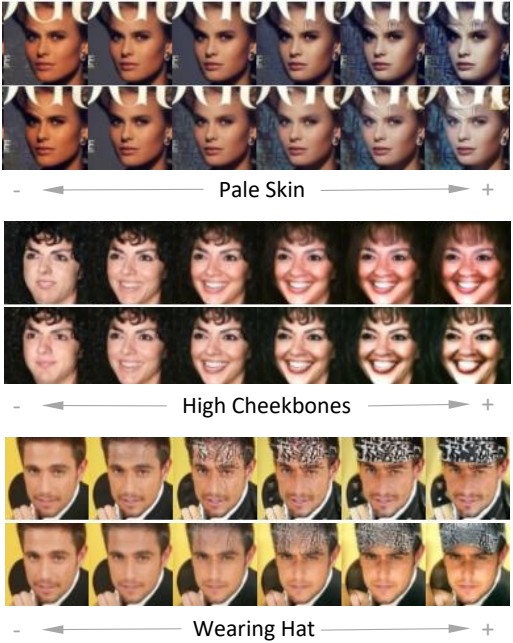

Figure E.7: Comparsions of PDAE (top row) and DiTi (bottom row)'s manipulation results on CelebA using all dimensions of the feature **z**.

PDAE even in this setting, *e.g.*, more meaningful background changes, less artifact and more convincing intermediate results.

**Interpolation Results on Bedroom**. In Figure E.6, we show the interpolation results on Bedroom, where our DiTi (bottom row) outperforms PDAE (top row) with less artifact. Note that Bedroom is especially challenging for interpolation due to the large variations between images. Larger model and prolonged training can benefit the interpolation fidelity. We leave it as future work as improving generation fidelity is not the focus of our work.

**Manipulation Results on CelebA**. We run the standard manipulation experiments (*i.e.*, without ProbMask) with results in Figure E.7. Our results (bottom row) are on par or better than PDAE (top row), *e.g.*, generating hat with less artifact. This experiment is to demonstrate that DiTi still has the conventional manipulation capability, and the aforementioned modular manipulation experiments are more suitable to highlight our disentanglement quality.

**PDAE Results Supplementary to Figure 5**. As shown in Figure E.8, the counterfactual generations by PDAE have more artifacts and are less meaningful compared to DiTi (Figure 5) on other feature subsets as well.

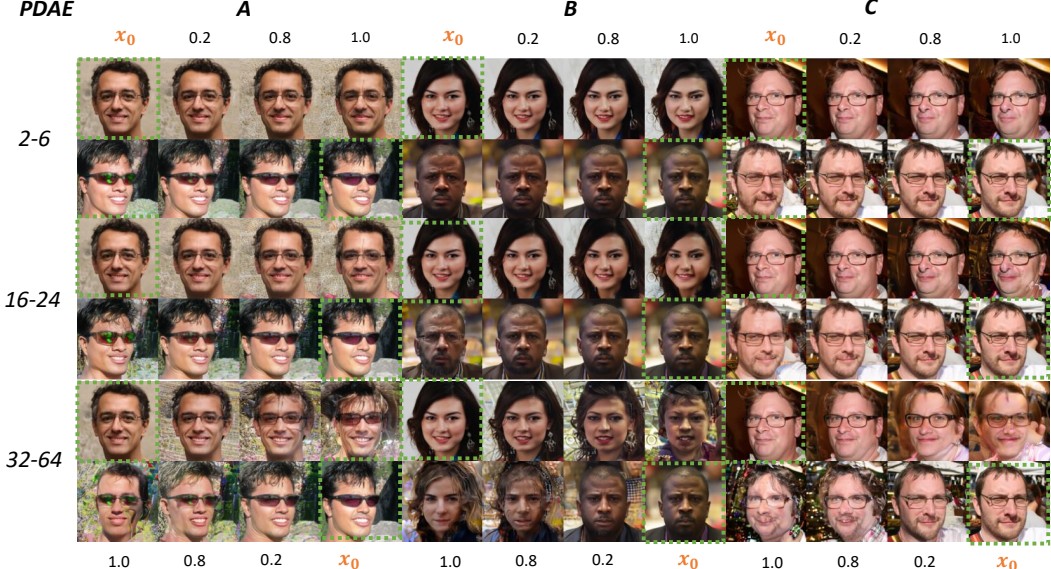

Figure E.8: Counterfactual generations results with PDAE on FFHQ. Supplementary to Figure 5 in the main paper.