# OpenReview forum: "Exploring Diffusion Time-steps for Unsupervised Representation Learning"
_ICLR.cc/2024/Conference — ICLR 2024 poster_

### Official Review · Reviewer_1YJJ · 2023-10-14

**Soundness:** 3 good
**Presentation:** 3 good
**Contribution:** 2 fair
**Rating:** 6
**Confidence:** 4

**Summary:**

I was one of the reviewers for this paper for NeurIPS 2023, I have carefully revisited the paper again for the ICLR submission, and write my new reviews as below.

This paper studies unsupervised representation learning with generative diffusion probabilistic models with specific exploration of the diffusion steps. The main research idea originates from the observation/assumption that image attributes are gradually lost along the diffusion process with increasing levels of Gaussian noises. The authors propose DiTi (named after Diffusion Time-step) to learn a step-specific feature to capture the information of lost attributes, and then leverage this feature of modular attributes as the inductive bias for unsupervised learning. Experimental tests of the feature are conducted on two settings, namely the attribute classification and counterfactual generation.

**Strengths:**

- The paper elaborates well on the concept of attribute loss, which is the fundamental observation for the proposed DiTi method, and is in general easy to follow.

- The idea to learn a step specific feature to encode the lost attribute information as the inductive bias for unsupervised learning is reasonable and intuitive.

- Experimental results show improvement over several baseline methods on CelebA, FFHQ and LSUN-Bedroom datasets.

- Compared to the previous manuscript from NeurIPS, it seems that the authors have incorporated the comparison of the proposed work with several existing works that touch on the representation learning with DMs, which helps to clarify the differences between this paper and existing literature (but this part is also a little biased, as specified below).

**Weaknesses:**

Since this is my second time reviewing this paper, I think most of my questions concerning the technical details have been resolved and clarified. While I acknowledge that the paper is well-written and has its merits to enlighten the studies of combining representation learning and diffusion models, there are a few of my concerns.

- First, I echo with my previous NeurIPS reviewers folks on the concern: whether it is the direct direction to actually integrate DMs into the current representation learning paradigm? Based on our previous discussions, the proposed method has a higher computational/time cost compared to its contrastive counterparts. And if the argument here is because the contrastive paradigms lose attribute information, then it seems to me the proposed DiTi may not be the optimal method to introduce the attribute inductive bias.

- Second, in the related work section, the authors include a comparison with the current DM for representation learning methods. I find this new part slightly biased, because the approaches in 2) and 3) are not really proposed under the context of representation learning. For instance, (Kwon et al., 2022) proposed the bottleneck feature of the U-Net under the context of image editing; similar to (Preechakul et al., 2022), in which separate autoencoders are learned to achieve editing purposes. I think while these works do touch the latent features/attributes in DMs, they tackle different scenarios other than representation learning. In other words, the first concern is not resolved by comparing these works.

- Going towards the technical level, one concern (less urgent in my sense) is that the proposed DiTi method shares quite similarities with PDAE, with the main difference in the disentanglement, with the cost of more hyper-parameters.

**Questions:**

In general, I don’t have specific questions to ask at this time and still have mixed feelings as in my first-time review of this work. I think this paper has its own merits/advantages and drawbacks/disadvantages at the same time, and thus keep my tentative rating as borderline but will update it according to other reviewers and further discussions later.

---

> ### Author Response · Authors · 2023-11-13
> **Author Response**
>
> Thanks for the in-depth comments and suggestions.
>
> **W1 DM for representation learning?**
> - Regarding computational cost, our DiTi takes slightly longer to train than SimCLR (40 hours vs 28 hours) on our evaluation datasets. However, this may not be the case in large-scale training, because 1) contrastively learning scales poorly w.r.t. data size, 2) our method distils knowledge from a pre-trained DM instead of learning from scratch.
> - More importantly, our theory shows that DM has a unique mechanism for representation learning: attributes of different granularities are separated along DM time-steps, where fine-grained attributes are lost earlier than coarse-grained ones. This serves as an effective inductive bias for learning a disentangled representation, enabling robust inference and counterfactual generation, and has huge potential that can be further explored in future, e.g. see Kc2k-W2, Cjby-Q2.
> - We also add that in general, learning representation with DM removes the need of data augmentation in contrastive learning, leading to several benefits: 1) As you pointed out, this helps preserve augmentation-related attributes. 2) Practitioners no longer needs to find the most suited augmentations, which would have been a daunting task given the heavy cost of contrastive learning especially in large-scale training. 3) This helps extend representation learning to modalities where data augmentations are not well-defined/work not as well, such as audio or other time-series data.
>
> **W2 Comparison with related work**. As you rightfully point out, these works tackle representation learning from the perspective of building a more versatile generative model, e.g., to enable image editing or interpolation. We include them in the related work because they essentially still pursue a good representation to enable image editing. As discussed in Section 3.1, a good representation should enable robust inference in downstream tasks, as well as image editing on a targeted attribute (i.e., counterfactual generation). We will revise related work to clarify this.
>
> **W3 Similarity with PDAE**. We clarify that the similarity with PDAE in implementation details (e.g., network design, hyper-parameters) is deliberate, such that emerged properties of disentangled representation are solely from our theory-inspired algorithm. We follow PDAE for its hyper-parameter choice, and our added hyper-parameter is solely to suit the need for disentangled representation learning, which has a clear-cut choice from our ablation study (Section 5.2).

---

> > ### Comment · Reviewer_1YJJ · 2023-11-18
> > **Thanks for the rebuttal**
> >
> > I appreciate the authors' efforts in rebuttal.
> > I think the clarity of this current version does improve a bit compared to my first-time review. While I stay prudent on the direction of DMs for representation in general, I acknowledge this work provides some insights into the exploration. Keeping my score as it is.

---

> > > ### Author Response · Authors · 2023-11-23
> > > **Thanks for the comments**
> > >
> > > Thank you again for the review and comments.

---

### Official Review · Reviewer_5NeX · 2023-10-30

**Soundness:** 3 good
**Presentation:** 3 good
**Contribution:** 3 good
**Rating:** 6
**Confidence:** 3

**Summary:**

This work proposes DiTi, a method that learns disentangled representations in an unsupervised fashion. The work first provides insights into the inductive bias of Denoising Diffusion Probabilistic Models that can be leveraged for learning decoupled features, with theoretical grounding. Then the work proposes to leverage the inherent connection between timesteps and modular attributes to learn a set of features from the residuals. Finally, the feature can be used for both downstream inference (e.g., attribute prediction) and counterfactual generation, surpassing prior works in both tasks.

**Strengths:**

* This work provides a link between the timesteps and the modular attributes with an intuitive explanation and theoretical proof.
* This work proposes a simple yet effective approach to learn disentangled features along with the diffusion model training that follows from the discoveries during the analysis.
* This work outperforms previous unsupervised feature learning methods that do not disentangle the features on attribute classification tasks. This work also shows higher-quality counterfactual generations compared to previous works.

**Weaknesses:**

* This work trains models from scratch. Due to the introduced term for disentangled feature learning, the model needs to be re-trained.
* The authors only evaluated the proposed method with datasets that are domain-specific (e.g., CelebA, FFHQ). The method has not been evaluated on large models trained on more diverse and general image datasets (e.g., LAION). Therefore, it does not indicate whether the method can scale to more complicated settings.

**Questions:**

* Do the subsets have to be uniform? Does it benefit the method if we have more features for coarser or finer features?

---

> ### Author Response · Authors · 2023-11-13
> **Author Response**
>
> Thanks for your review. We address all weaknesses and questions below.
>
> **W1 Model Training from scratch**. We clarify that we only train the feature extractor from scratch, which is standard in representation learning. We use a pre-trained DM and freeze it in representation learning. Our training cost is similar with the recent related work PDAE and much faster compared to previous works, e.g., Diff-AE. Could you kindly clarify what you mean by “the model needs to be re-trained”?
>
> **W2 Training on LAION**. We follow the standard evaluation datasets in this domain following PDAE and Diff-AE. Due to limited GPU access at the point of submission (4$\times$A100), it is too expensive to train on LAION. However, our method is grounded on a provable theory, which still holds on more diverse datasets. In fact, we are currently doing a large-scale pre-training based on this method.
>
> **Q1 Non-uniform Subsets**. Yes, our method benefits by having more dimensions on the feature subset corresponding to time-step range 100-300. Please refer to the implementation details in Section 4.2, the ablation results in Section 5.2 and additional details in Appendix Section E.

---

> ### Comment · Reviewer_5NeX · 2023-11-21
>
> Thanks for the rebuttal. I still recommend acceptance of this work, as reflected in my rating.
>
> As a clarification for W1 and W2, different from works that take information from pretrained diffusion models that only require a very small amount of compute workload to realize or improve their downstream (e.g., [1]), this work presents another term in the loss for better representation learning that contributes to improved downstream, which makes it need to be re-trained (i.e., not using the information from the pre-trained models from large datasets such as LAION). The reviewer accepts the theoretical foundations for this work, but the need for retraining still leads to barriers to users who do not have the compute to train the representation on a large dataset but have access to public weights of diffusion models. The reviewer is glad that the authors are training a model on large datasets with the method presented in their work and encourages the public release of the trained model.
>
> [1] Diffusion Hyperfeatures: Searching Through Time and Space for Semantic Correspondence. NeurIPS 2023. https://arxiv.org/abs/2305.14334

---

> > ### Author Response · Authors · 2023-11-23
> > **Thanks for the comments**
> >
> > Thank you for the clarification. We agree & highlight that representation learning in general requires a large amount of compute and we strive to release an open-source model trained on larger dataset in future.

---

### Official Review · Reviewer_Cjby · 2023-10-31

**Soundness:** 3 good
**Presentation:** 3 good
**Contribution:** 3 good
**Rating:** 6
**Confidence:** 5

**Summary:**

This work aims to disentangle the modular attributes in the DDPM framework by exploiting the granularity of feature details at different time steps of the diffusion models. The work finds that the fine feature components are encoded in the earlier time steps of the forward diffusion process while the coarse details are at the later time steps. A encoder decoder approach with partitioned features  at different time steps are proposed to allow for disentangled representations. Experiments are performed on bedroom, CelebA and FFHQ datasets where the method outperforms prior art.

**Strengths:**

+ The paper is well-written and easy to read. The motivation of the work related to learning the disentangled modular attributes in diffusion models is promising. Claims are supported with theoretical reasoning.

+ The experiments are performed on different face datasets such as CelebA, FFHQ, and the Bedrooms dataset. The approach outperforms prior work on metrics such as AP, Pearson-r, and MSE.

+ Ablations are provided regarding the choice of partitioning and optimization strategies.

**Weaknesses:**

- The qualitative results shown in figure 5 do not show consistent improvements wrt to modular editing. In the second row, for example, the eyeglasses appear in the middle range and disappears again in the final ranges.  Are the time steps plotted in a cumulative fashion within the  early/middle/late “t” range.

- On the use of timesteps [100-300] and it corresponding to the maximum value of the loss: One would expect the loss to be greater at the later stages when the image becomes complete noise. Eg: it should be difficult to construct image from t =1000 to t =700. Can a plot be included for different samples to validate this?

- Comparison to SIMCLR: SimCLR does obtain a higher accuracy on attributes affecting local appearances (e.g., “Hat”). We postulate that its contrastive training has some effects in regularizing f as an injective mapping” . This is not clear. What is being implied in these findings. Does this mean that the proposed approach depends on attribute correlations?

**Questions:**

- From point 2 above, it will be great to include the plot to validate the time steps and loss.
-The results presented in the manuscript hold for unconditional models. Do the findings also extend to conditional models?
Also see weaknesses above.


Minor: The equations are missing parenthesis on expectation.

---

> ### Author Response · Authors · 2023-11-13
> **Author Response**
>
> Thanks for the in-depth comments and suggestions. We will fix the typo and address all points below.
>
> **W1 Modular editing consistency**. No, we separately interpolate each feature subset corresponding to early/middle/late time-step range, as indicated on top of Figure 5. This is why we expect the edit of eyeglasses to happen in one range instead of multiple ranges. For example, if interpolating the feature subset in the middle range changes the “eyeglasses” attribute, interpolating the subset in the late range should only modify a more coarse-grained attribute without changing “eyeglasses”.
>
> **W2 & Q1 Plot of loss against time-steps**. We include the plot in Appendix Figure A9. The reason for a small loss at a large time-step $t$ is that DM assigns a weight to each time-step (Eq. 2), and the weight is extremely small at a large $t$. One could also understand this from the $\epsilon$-formulation (equivalent to our reconstruction formulation), where the U-Net predicts the added noise in the input $x_t$. At a large $t$, $x_t$ is close to a pure noise, hence a trivial prediction by directly outputting $x_t$ already achieves a small loss.
>
> **W3 Comparison with SimCLR**. Sorry for the confusion. We are not suggesting that our proposed DiTi relies on attribute correlation. Instead, we are suggesting possible reasons that SimCLR outperforms our DiTi on the classification accuracy of a few attributes (blue bars in Figure 4). SimCLR’s contrastive objective encourages different samples to have different features (i.e., representation must be an injective mapping). Hence the representation is encouraged to account for all differences between each sample pair to make their features maximally contrastive. This can help representation capture unnoticeable attributes (e.g., earrings). It may be worth exploring in future the merit of contrastive learning in context of diffusion models, e.g., as in [1].
>
> [1] Discrete Contrastive Diffusion for Cross-Modal Music and Image Generation. ICLR 2023
>
> **Q2 Extension to conditional models**. Good suggestion. We did some follow-up experiments to verify our theory on a text-conditioned DM. Specifically, our theory indicates that only fine-grained attributes are lost at a small time-step $t$. Hence given images from a fine-grained category (e.g., a specific aircraft model), we first obtain their noisy images at a small $t$, then we fine-tune a text-conditioned DM (Stable Diffusion) to reconstruct the original images from noisy ones conditioned on a chosen prompt (e.g., a rare token). If the reconstruction is accurate, the finetuned DM must make up for the attribute loss by associating the prompt with the category’s fine-grained attribute. In this way, to classify if a test image $x_0$ is from this category, we can intervene on the diffusion time-step to only focus on the fine-grained attribute, i.e., if DM can reconstruct it from $x_t$ at a small $t$, then we predict as “yes”, because the fine-grained attribute of $x_0$ matches that of this category. We verify that this intervention strategy on diffusion time-step is extremely effective in removing spurious correlations in the difficult few-shot learning task, e.g., on the challenging FGVC-Aircraft dataset, we significantly outperform previous few-shot learning methods using OpenCLIP (ViT-H/14 trained on LAION-2B) by up to 13%.
>
> Table r3: $N$-shot performance on FGVC-Aircraft dataset.
> | Method                      | 1        | 2        | 4        | 8        | 16       | Avg      |
> | --------------------------- | -------- | -------- | -------- | -------- | -------- | -------- |
> | Zero-Shot OpenCLIP          | 42.3     | 42.3     | 42.3     | 42.3     | 42.3     | 42.3     |
> | Zero-Shot Stable Diffusion  | 24.3     | 24.3     | 24.3     | 24.3     | 24.3     | 24.3     |
> | Tip-adapter (on OpenCLIP)           | 48.4     | 53.9     | 57.0     | 62.0     | 67.4     | 57.7     |
> | **DiTi ( on Stable Diffusion)** | **48.5** | **55.8** | **64.2** | **74.2** | **79.9** | **64.5** |

---

> > ### Comment · Reviewer_Cjby · 2023-11-22
> > **Thank you for the rebuttal**
> >
> > I am satisfied with the author's rebuttal for W2 and W4.
> > For W1, Figure 5 still does not clarify the modular editing. So according to the rebuttal, there should have been consistent behavior regarding the editing of the eyeglasses for pairs 2 and 3 in either direction. This is not consistent.  The reason for looking at these particular examples is that it is easy to evaluate the consistency of the approach visually.
> >
> > For W3, I would urge the authors to add clarifications and revisions to the text in the paper.
> >
> > I will keep my score.
> >
> > Thanks!

---

> > > ### Author Response · Authors · 2023-11-23
> > > **Thank you for the comments**
> > >
> > > Thank you for the review and comments. We have added the clarification for W3 and parenthesis on expectation in revision.
> > > For Figure 5, we believe the editing is consistent, as from left (w/ sunglasses) to right (w/o sunglasses), the editing should be adding sunglasses (Figure top), and from right to left, the editing should be removing sunglasses (Figure bottom).

---

> > > > ### Comment · Reviewer_Cjby · 2023-11-23
> > > > **Clarification**
> > > >
> > > > For figure 5, In the figure on the top looks good in the middle range, however for the bottom figure in the middle range, the glasses still appear, it will be great if this can be clarified. Thanks for making revisions to the paper.

---

> > > > > ### Author Response · Authors · 2023-11-23
> > > > > **Clarifying Figure 5**
> > > > >
> > > > > Thanks for the prompt response. For each feature range, we visualise a trajectory of three generated images by gradually increasing the interpolation scale (illustrated by the colour bars on top of images). Note that the direction is different for the top three rows and bottom three rows: For the top, we interpolate from left to right, gradually adding sunglasses, and the **rightmost** figure completely adds sunglasses to the person. For the bottom, we interpolate from right to left, gradually removing sunglasses, and the **leftmost** figure completely removes the sunglasses.

---

### Official Review · Reviewer_Kc2k · 2023-11-04

**Soundness:** 3 good
**Presentation:** 3 good
**Contribution:** 2 fair
**Rating:** 6
**Confidence:** 3

**Summary:**

This paper presents a unique perspective on exploiting the image-noise ratio across different time steps in diffusion models and offers a framework for learning useful and disentangled representations from diffusion models. The authors argue that as t increases, images progressively loss information starting from details to global structures during noise injection, and learning a complementary feature representation conveying the corrupted information can help make the representation more disentangled. Experiments are performed on multiple image datasets.

**Strengths:**

1. The idea of systematically studying the representation changes during the noise injection of diffusion models is interesting and has great potential in further understanding the diffusion models.

2. The paper is written in good presentation quality in general. The definitions such as attribute loss are adequately formulated and illustrated with figures.

3. The quantitative results in Table 1 seem strong.

4. The method is built on top of pretrained diffusion models, which I believe can help reduce the cost of representation learning.

**Weaknesses:**

The empirical discussions on counterfactual generation (section 5.3) are relatively weak as the effectiveness is only supported by a few examples. Especially for the bedroom experiments, I personally feel like it's not easy to justify the authors' claim that 'DiTi is the only method that generates faithful counterfactuals' based on the given examples.

And since this paper focuses on unsupervised representation learning, more empirical results regarding how the learned disentanglement can improve the overall representation quality can be a huge plus.

For example, do the learned disentangled representations enable human intervention to mitigate spurious correlations?

How is the quality of the representation when evaluated in a more general setting such as ImageNet-style recognition?

One concern I do have is that as diffusion is trained by image reconstruction, the learned feature will inevitably contain considerable information regarding the background (probably in late step t), and how will this affect the general representation quality?

**Questions:**

Could the authors provide a brief discussion for [1]?


---
[1] Diffusion Based Representation Learning, arxiv

---

> ### Author Response · Authors · 2023-11-13
> **Author Response #1**
>
> Thank you for the detailed review and insightful suggestions.
>
> **W1 Counterfactual generation results**. We are sorry that our presentations might have confused you. We believe that our counterfactual generation results are sufficient to support that our DiTi learns a more disentangled representation compared to baselines. Specifically, we have two main types of experiments: 1) By interpolating a feature subset between a pair of images (Figure 5), and 2) by manipulating a feature subset guided by the weight of an attribute classifier.
> - For 1), only our DiTi allows meaningful interpolation between image pairs as shown in Figure 5 and A11 in Appendix. Hence, we include the baseline results in Appendix (Figure A12-A13), where the generated counterfactuals are either identical to the original image, or contain substantial artifacts, validating the baseline features are still entangled.
> - For 2), a disentangled representation should enable manipulating a single attribute without affecting other ones. As shown in Figure 2(a), 6 and A14-A16 in Appendix, baseline methods typically entangle multiple attributes (e.g., manipulating age and gender together), and are still prone to generating visual artifacts. In contrast, our DiTi mainly manipulate only the target attribute.  Note that the results can be quite subtle when the attribute is fine-grained, e.g., “wearing lipstick”. Following your suggestion, we will adjust the claim about bedroom experiments to reflect the aforementioned point.
> - We have other supporting experiments to verify disentanglement quality, e.g., visualization of classifier weight (explained in the last paragraph of Section 3.1), strong inference results as noted by other reviewer (Section 5.2), and the results of interpolating the whole feature instead of a feature subset (explained in the last paragraph of Section 5.3).
> Overall, our results are from multiple evaluation methods, on different datasets, with many examples, and we also compare with diverse baselines (e.g., including conventional disentanglement methods based on VAE and GAN). Hence we believe that they are sufficient to validate our disentanglement quality.
>
> **W2 Mitigating spurious correlation by disentangled representation**. Great suggestion. We evaluate our method and baselines on a few-shot learning task for the fine-grained attributes in CelebA. This is particularly challenging, as the classifier needs to isolate fine-grained class attributes from visually prominent yet spurious ones. As our method disentangles fine-grained attributes on the feature subset corresponding to a small $t$ (e.g., early $t$ in Figure 5), we can perform human intervention to only use that subset to train a classifier, which discards the spurious attributes. As shown in Table r1 and r2, using the full feature of DiTi for few-shot learning does not bring much improvements, as the classifier can still exploit the spurious correlations on the disentangled coarse-grained attributes. However, after intervention, we observe significant improvements over baselines.
>
> Table r1: $N$-shot accuracy on "smiling" attribute averaged over 10 runs.
> | Method            | 5    | 10   | 20   | 50   |
> | ----------------- | ---- | ---- | ---- | ---- |
> | Diff-AE           | 56.1 | 60.2 | 62.2 | 67.8 |
> | PDAE              | 55.5 | 59.5 | 61.8 | 66.5 |
> | DiTi              | 56.0 | 60.4 | 62.6 | 68.4 |
> | DiTi+intervention | 58.5 | 62.7 | 67.4 | 71.8 |
>
> Table r2: $N$-shot accuracy on "wearing lipstick" attribute averaged over 10 runs.
> | Method            | 5    | 10   | 20   | 50   |
> | ----------------- | ---- | ---- | ---- | ---- |
> | Diff-AE           | 55.2 | 57.4 | 60.3 | 62.3 |
> | PDAE              | 55.0 | 56.6 | 59.7 | 62.1 |
> | DiTi              | 55.3 | 57.1 | 60.2 | 62.7 |
> | DiTi+intervention | 56.5 | 59.2 | 62.4 | 65.6 |
>
> In fact, we have performed similar follow-up experiments using a text-conditioned DM (Stable Diffusion), where our method beats few-shot learning methods using OpenCLIP by up to 13%. See Cjby-Q2 for details.

---

> ### Author Response · Authors · 2023-11-13
> **Author Response #2**
>
> **W3 ImageNet-style recognition**. Unfortunately, as our representation is trained on face or bedroom dataset, it cannot be directly used for ImageNet evaluation. As we have limited GPU access at the point of submission, it would also be too expensive to train a representation on ImageNet. However, our theory and method can be extended to a larger and more diverse dataset, and we are currently working on large-scale pre-training with this method.
>
> **W4 Background information**. As rightfully pointed out, our DiTi will disentangle background information in the feature subset corresponding to a late time-step $t$. However, we highlight that this is in fact desirable, because:
> 1) The background information may be useful in downstream tasks, e.g., in semantic segmentation on road images, the model needs to classify the type of each background pixel into road, pavement, etc.
> 2) If background is irrelevant for a downstream task, one can perform human intervention to discard the feature subset corresponding to late time-steps, such as the above few-shot learning examples on fine-grained categories.
>
> **Q1 Discussion for [1]**. The paper shares a similar high-level idea, i.e., learning a feature as an input condition for DM to enable perfect reconstruction. Key differences: 1) [1] relies on empirical evidences and hypothesis for the concept of attribute loss, while we provide a rigorous theoretical analysis. 2) [1] learns an infinite-dimension feature, while we learn a compact feature that is easy for downstream leverage.

---

### Meta-Review · Area_Chair_U8Fa · 2023-12-05

**Metareview:**

This paper addresses the problem of learning disentangled representations using diffusion models. The idea is that different types of features are learnt at different timesteps in diffusion, and hence an encoder decoder approach is proposed with partitioned features at different time steps to allow for disentangled representations. This idea is quite intuitive and interesting to study in the context of unsupervised disentangled representation learning. The paper is also well written and it is easy to understand.

Most of the reviewers like this idea and lean towards accepting the paper. Main concerns include lack of clarity in some sections, missing large scale experiments. I believe the authors have improved the clarity of some sections in rebuttal. Regarding large scale experiments, it is true that they are missing. But I feel sufficient results are shown on small-scale benchmarks such as CelebA and FFHQ. Experimental results are also quite good.

Reviewer 1YJJ raised a concern that if it is indeed a right direction to use DMs for representation learning. While this is an interesting debate in itself, I feel this paper has enough contributions and this study can help the community understand this debate better. It is true however, there are some cons with this approach such as increased inference time. But, in my opinion, the contributions outweigh the issues, and hence I vote for accepting this paper.

**Justification For Why Not Higher Score:**

While the idea is interesting and the paper is well written, I feel the contribution is not up to the level of a oral paper. There are some challenges associated with this approch such as slow inference speed that can make this a general solution for representation learning. Experiments on large-scale would have made this paper even stronger. Most of the results are good, but they are not super impressive.

**Justification For Why Not Lower Score:**

The paper has sufficient contributions. The idea of using DMs for disentangled representation learning is an interesting one to study. Experiments sufficiently validate the usefullness of this approach.

---

### Decision · Program_Chairs · 2024-01-16

Accept (poster)